https://doi.org/10.1038/s42003-023-05737-7　　**OPEN**
# Mendelian randomization identifies circulating proteins as biomarkers for age at menarche and age at natural menopause

Nahid Yazdanpanah[1,5], Basile Jumentier[1,5], Mojgan Yazdanpanah[1,5], Ken K. Ong [2], John R. B. Perry [2,3] & Despoina Manousaki [1,4 ✉]

Age at menarche (AAM) and age at natural menopause (ANM) are highly heritable traits and have been linked to various health outcomes. We aimed to identify circulating proteins associated with altered ANM and AAM using an unbiased two-sample Mendelian randomization (MR) and colocalization approach. By testing causal effects of 1,271 proteins on AAM, we identified 22 proteins causally associated with AAM in MR, among which 13 proteins (GCKR, FOXO3, SEMA3G, PATE4, AZGP1, NEGR1, LHB, DLK1, ANXA2, YWHAB, DNAJB12, RMDN1 and HPGDS) colocalized. Among 1,349 proteins tested for causal association with ANM using MR, we identified 19 causal proteins among which 7 proteins (CPNE1, TYMP, DNER, ADAMTS13, LCT, ARL and PLXNA1) colocalized. Follow-up pathway and gene enrichment analyses demonstrated links between AAM-related proteins and obesity and diabetes, and between AAM and ANM-related proteins and various types of cancer. In conclusion, we identified proteomic signatures of reproductive ageing in women, highlighting biological processes at both ends of the reproductive lifespan.

[1] Research Center of the Sainte-Justine University Hospital, University of Montreal, Montreal, Quebec, Canada. [2] MRC Epidemiology Unit, Wellcome-MRC Institute of Metabolic Science, University of Cambridge School of Clinical Medicine, Cambridge CB2 0QQ, UK. [3] Metabolic Research Laboratory, Wellcome-MRC Institute of Metabolic Science, University of Cambridge School of Clinical Medicine, Cambridge CB2 0QQ, UK. [4] Departments of Pediatrics, Biochemistry and Molecular Medicine, University of Montreal, Montreal, Canada. [5] These authors contributed equally: Nahid Yazdanpanah, Basile Jumentier, Mojgan Yazdanpanah. ✉email: despina.manousaki@umontreal.ca

Menarche and menopause are two major events in a woman's reproductive life.
Growing evidence indicates that early or late occurrences of these events, as part of their natural age variation in the general population, are associated with increased risks of adverse health conditions such as endometriosis, breast cancer, adult obesity, type 2 diabetes, cardiovascular disease, and increased mortality[1]. In an effort to prevent these adverse health effects, hormonal-blocking or replacement treatments aim to directly rectify the physiological changes caused by extreme variation in ages at menarche or menopause. Understanding the pathways underlying normal pubertal and menopausal timing in women is paramount to differentiate between natural variations in the reproductive lifespan versus changes due to pathological causes. This knowledge can guide medical decisions to treat women with earlier or later occurrences of menarche or menopause to mitigate related long-term adverse effects.

In the majority of cases, premature or delayed menarche or menopause are extreme presentations of the normal physiological variation in the timing of puberty or reproductive aging[2]. Simple hormonal measurements in the blood often cannot help differentiate between these normal variations and epiphenomena of underlying pathological processes. Thus, there is an unmet need for discovery of predictive biomarkers of extreme physiological variations in the age of female reproductive aging, while the same molecules could be targets for pharmacological treatment. The large amount of data generated in high throughput proteomic studies offers a source to comprehensively study differences in levels of circulating proteins associated with variation in age at menarche (AAM) and age at natural menopause (ANM), which individually, or as part of a network, can provide insight into the molecular mechanisms that affect timing of menarche and of natural menopause[3,4]. Circulating proteins in the blood have been a valuable source for biomarker discovery and drug target characterisation for various diseases[5]. However, the serum proteome can be highly sensitive to sample processing, and its measurement is costly and prone to potential bias due to differences in the sample groups. Case-control studies comparing proteomic profiles between individuals affected with diseases versus controls can be subject to reverse causation, but if our goal is prevention, it is necessary to detect changes in proteomic profiles prior to disease onset. Due to the above limitations, it is complicated to comprehensively characterize and compare the serum/plasma proteome in case-control studies[6]. Better evidence is needed to understand if circulating proteins play a causal role in female reproductive aging.

In the recent years, large-scale proteomic GWAS have identified thousands of variants controlling levels of circulating proteins[7–11]. Also, AAM and ANM are highly heritable traits with estimates of heritability for both AAM and ANM varying between 50% and 80%[12–15]. These estimates have been confirmed by recent large-scale genome-wide association studies (GWAS) on AAM and ANM, leveraging data from over a million women[16–18]. These GWAS provide a valuable opportunity to utilize Mendelian randomization (MR), a method allowing to test causal associations between modifiable exposures (such as circulating proteins) and outcomes, such as AAM and ANM. Under certain assumptions, MR uses genetic variants as instruments to estimate the effect of an exposure on an outcome of interest[19], exploiting the random allocation of these variants at conception to infer causal effects that are robust to confounding[19] and reverse causation, which are major limitations of conventional observational studies[20].

In this study, we aimed to leverage data from large proteomic GWAS, to identify circulating proteins, genetically predicted levels of which are associated with AAM or ANM within the MR framework. To further prioritize the MR-identified proteins, we undertook colocalization, followed by pathway, enrichment analyses and analyses using expression profiles.

## Results

To evaluate the causal role of circulating proteins on AAM, and ANM, we identified *cis*-pQTLs from six proteomic GWAS (Sun et al.[7], $N = 3301$; Emilsson et al.[8] $N = 3200$; Suhre et al.[9], $N = 1000$; Folkersen et al.[10], $N = 21{,}758$; Yao et al.[11], $N = 6861$, and Ferkingstad et al.[21], $N = 35{,}559$) and used them as genetic instruments (Fig. 1). To assess the association of *cis*-pQTL with

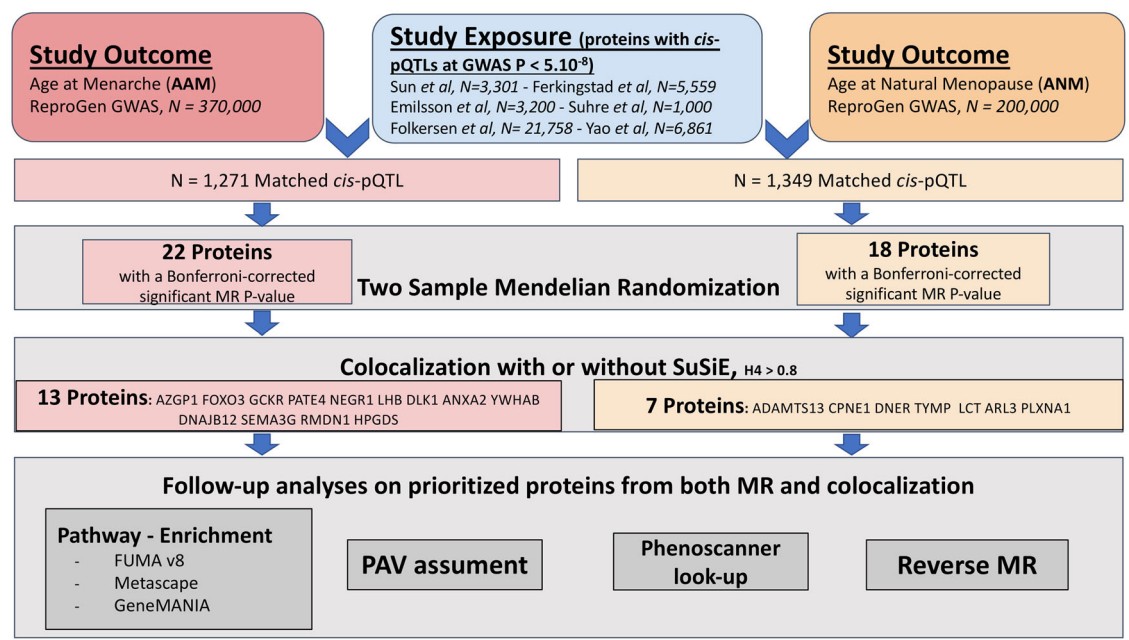

**Fig. 1 Flowchart with the study design.** Mendelian Randomization was carried out with the R package TwoSampleMR. Colocalization was carried out with the R package coloc and the SuSIE plug-in.

AAM, we retrieved the effects of these *cis*-pQTL on AAM in 370,000 European women from the REPROGEN Consortium GWAS[16] (Supplementary Data 1). We obtained effects of the *cis*-pQTLs on ANM from the largest GWAS meta-analysis on ANM equally from the REPROGEN Consortium[18], totaling 200,000 women of European descent from 21 studies. (Supplementary Data 1).

The results of the MR analysis, expressing changes in AAM or ANM in years per 1 standard deviation (SD) increase in the levels of a circulating protein, are presented in Fig. 2. Our power analysis[22] showed that at an alpha = 0.05/1,300 = 3.8 x 10$^{-5}$ (after Bonferroni correction), our power to detect an effect as small as 0.02 years in AAM or of 0.11 years in ANM per SD change in a specific protein level was 100% for the majority of the MR-prioritized proteins, based on the variance explained of each protein by its SNP-instrument, and a reported SD for AAM of 1.3 years and for ANM of 4 years (Supplementary Data 2 and 3).

**Causal effects of circulating proteins on AAM.** In total, 1271 distinct proteins (instrumented by 1265 directly matched *cis*-pQTL and 6 proxies) were tested for association with AAM using MR. Considering Bonferroni correction (*P* = 0.05/1271 or 3.9 x 10$^{-5}$, Supplementary Data 4), we identified 22 proteins associated with AAM. The *cis*-pQTL of the 22 proteins had all an

F-statistic > 10, ranging from 35.12 (NEGR1) to 25208.6 (TXNDC15), explaining from 0.1% (FOXO3) to 77.1% (MST1) of the variance of the protein levels. The 22 proteins presented MR beta coefficients (β) on AAM ranging from -0.45 (FOXO3) to 0.46 (MANF) years per 1 SD increase in protein levels (Fig. 2a, Supplementary Data 5).

We cross-validated the findings for LHB and TIE1 levels, for which *cis*-pQTLs effects were available in both Sun et al.[7] and Emilsson et al.[8], and the MR effect sizes on AAM were comparable (Fig. 2a). Similarly, the result using a *cis*-pQTL for YWHAB in Ferkingstad et al.[21] was replicated using a *cis*-pQTL from Emilsson et al.[8], the finding for MST1 using a *cis*-pQTL from Suhre et al.[9] was replicated using a *cis*-pQTL from Emilsson et al.[8] (Fig. 2a), and the MR effect of MANF using a *cis*-pQTL from Ferkingstad et al.[21] was consistent with that using a *cis*-pQTL from Emilsson et al.[8] (Supplementary Data 4). The *cis*-pQTL associated with the cross-validated proteins differed between the two proteomic GWAS-sources for all of the 5 aforementioned proteins.

To assess if our MR results were driven by confounding due to linkage disequilibrium (LD), we tested whether the MR-prioritized proteins shared a single causal variant with AAM using colocalization, where a posterior probability 4 (H4) indicates presence of a single causal SNP for the protein and AAM, while a posterior probability 3 (H3) indicated the presence

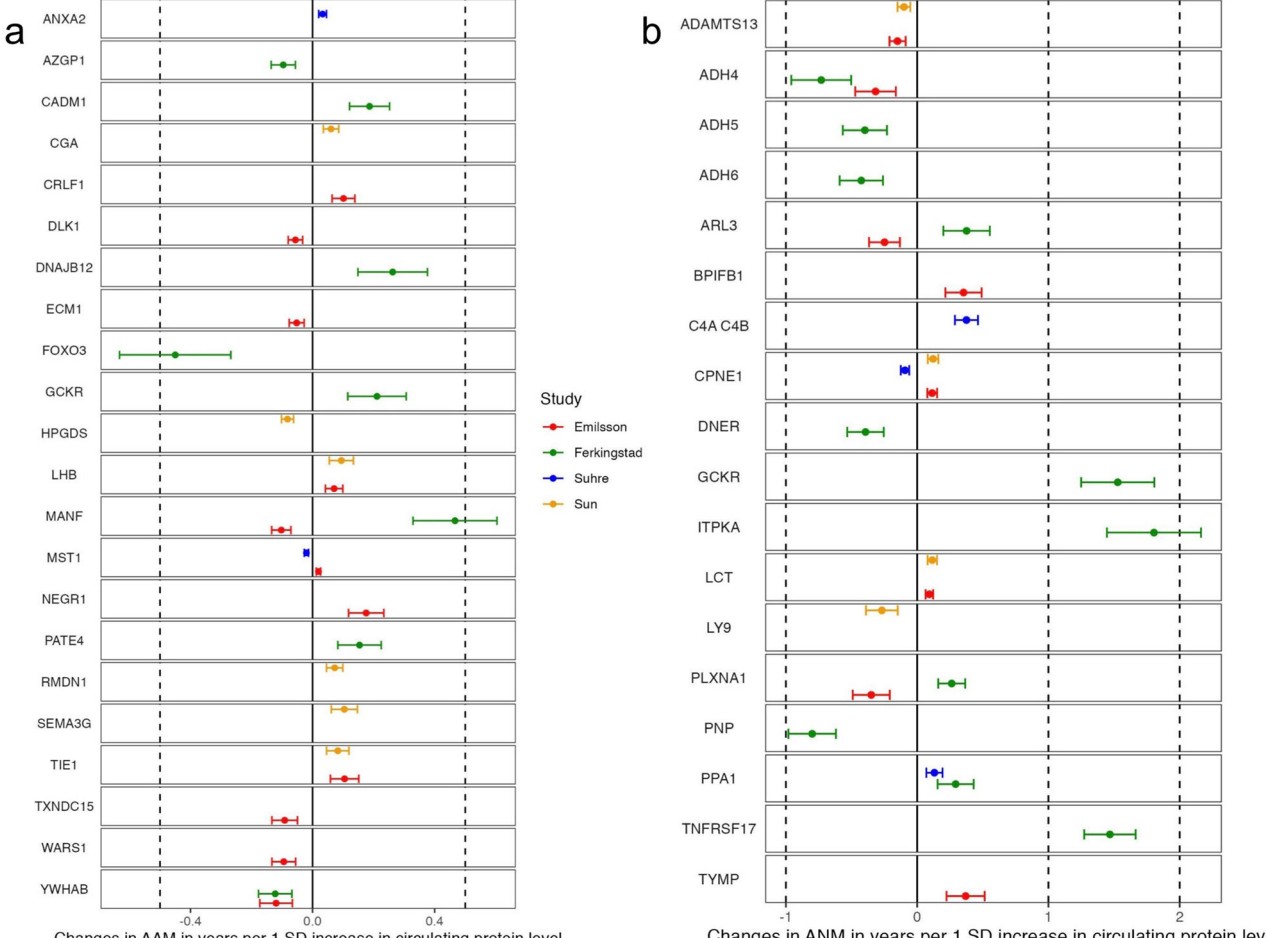

**Fig. 2 MR estimates for 22 proteins associated with AAM, and 18 proteins associated with ANM in our MR analysis.** Forest plots show the association between genetically determined level of each protein with AAM or ANM using MR. β coefficients represent the effect of a 1 SD higher protein level derived from the genetic instrument on the respective AAM, and ANM trait measured in years. **a** Biomarkers associated with AAM. **b** Biomarkers associated with ANM. See Supplementary Data 4 and 9 for more information. †*P* = 0.05/1,271 or 3.9 x 10$^{-5}$; ‡*P* < 3.7 × 10$^{-5}$ (0.05/1349).

of two causal variants (one for the protein and the other for AAM) in LD. For 18 of the 22 MR-prioritized proteins, summary-level GWAS results were publicly available in the Sun et al.[7], Suhre et al.[9], or Ferkingstad et al.[21] GWAS, allowing colocalization analyses. Five proteins [GCKR (*cis*-pQTL: rs1260326, H4 = 97%), FOXO3 (*cis*-pQTL : rs3813498, H4 = 95%), SEMA3G (*cis*-pQTL rs2016575, H4 = 92%), PATE4 (*cis*-pQTL rs665677, H4 = 89%) and AZGP1 (*cis*-pQTL rs61381915, H4 = 83%)] showed high evidence of colocalization with AAM with a > 80% posterior probability H4 of sharing a single causal variant with AAM. We found lower evidence of colocalization, but with the H4 being the greatest among all the posterior probabilities, for DBAJB12 (*cis*-pQTL: rs9416018, H4 = 77%), HPGDS (*cis*-pQTLrs1965049, H4 = 66%), LHB (*cis*-pQTL rs79502742, H4 = 59%) and YWHAB (*cis*-pQTL rs6031848, H4 = 53%) (Supplementary Data 6). The remaining 9 proteins among the 18 tested for colocalization showed evidence of association with AAM through two different causative variants in LD (all H3> 93%) (Supplementary Data 6). Using the Sum of Single Effects (SuSiE) plug-in in colocalization[23], we found that 8 additional proteins (NEGR1, LHB, DLK1, ANXA2, YWHAB, DNAJB12, RMDN1 and HPGDS) colocalized with AAM with an H4 > 80% if we considered multiple causal variants within a 1 Mb region of each protein's *cis*-pQTL (Supplementary Data 7).

**Causal effects of circulating proteins on ANM.** We used 1349 *cis*-pQTLs as genetic instruments to test for causal associations of an equal number of circulating proteins with ANM. Considering Bonferroni correction ($P = 0.05/1349$ or $3.7 \times 10^{-5}$), we identified 18 proteins significantly associated with ANM, with MR beta coefficients (β) ranging from −0.80 (PNP) to 1.80 years (ITPKA) per 1 SD increase of protein levels (Fig. 2b). All 19 *cis*-pQTL of the candidate proteins had F-statistics > 10 ranging from 47.4 (TYMP) to 1273.3 (CPNE1), and explained between 0.2% (ITPKA) and 28.5% (CPNE1) of the variance in the protein levels (Supplementary Data 8). Furthermore, we cross-validated the findings for 4 proteins (LCT, PPA1, ARL3, PLXNA1) in at least two GWAS studies except for CPNE1 which replicated across 3 proteomic GWAS studies (Supplementary Data 9). The *cis*-pQTL for LCT and PPA1 were identical in the two proteomic GWAS-sources, but for the three other cross-validated proteins (ARL3, PPA1 and CPNE1), the *cis*-pQTL differed.

Next, we performed colocalization analyses for ANM and the 19 candidate proteins using summary-level results from Sun et al.[7], Suhre et al.[9] or Ferkingstad et al.[21]. We found evidence of colocalization for four proteins: CPNE1 (*cis*-pQTL: rs12481228, H4=100%) TYMP (*cis*-pQTL: rs131805, H4=98.4%), DNER (*cis*-pQTL: rs34412673, H4=97.2%), and ADAMTS13 (*cis*-pQTL: rs71503194, H4=91.3%). Our analysis indicated that 12 MR-prioritized proteins were associated with ANM through two causal variants in LD (H3> 80%) (Supplementary Data 10). Using colocalization with SuSiE[23], we found that 3 additional proteins (LCT, ARL and PLXNA1) colocalized with ANM with a posterior probability H4 > 0.8 if we considered multiple causal variants within a 1Mb region of each protein's *cis*-pQTL (Supplementary Data 11).

By comparing the MR-prioritized proteins of the two studied traits, we observed that GCKR (*cis*-pQTL: rs1260326 in Ferkingstad et al.[21]), which showed an increasing effect on ANM in our MR analysis, demonstrated an effect in the same direction on AAM. By using the R package *HyPrColoc*[24] we did not find evidence that GCKR colocalized simultaneously with AAM and ANM (Supplementary Data 12). This suggests that GCKR levels could predict later age at both natural menopause and menarche.

**Reverse MR analyses.** In order to test the presence of reverse causation in the association of the candidate proteins for AAM and ANM, we first performed a Steiger directionality test for each protein MR. We obtained a 'TRUE' results in all Steiger tests, which confirmed that the direction of causality was from the protein to AAM or ANM and not the opposite (Supplementary Data 13 and 14). To further explore the direction of the significant MR associations, we performed reverse MR studies, using AAM or ANM as exposures and the candidate proteins as outcomes. For these analyses, we used 172 and 193 genome-wide significant (GWAS *p*-value $< 5 \times 10^{-8}$) and independent (LD $R^2 < 0.001$) SNPs as instruments for AAM and ANM respectively, which we retrieved from the same REPROGEN Consortium GWAS described above. Our reverse MR analyses were restricted to proteins with *cis*-pQTLs identified in Sun et al, Suhre et al, and Ferkingstad et al. since full summary-level results of these GWAS were available. We tested reverse MRs for 20 out of the 22 candidate proteins for AAM and for all 19 candidate proteins for ANM. The results of these analyses appear in Supplementary Data 15 (for AAM) and 16 (for ANM). For each reverse MR analysis, we computed MR estimates using the inverse variance weighted (IVW) method, and three other pleiotropy-robust methods (MR-Egger, weighted median and weighted mode). For ANM, our reverse MR showed evidence of association (MR *p*-values < 0.05) in one out of the four MR methods for 6 proteins (DLK1, DBAJB12, GCKR, NEGR1, TXNDC15, MST1), but for LHB we found stronger evidence of a reverse causal effect of AAM on the level of this protein, with significant results in three out of the four MR methods (Supplementary Data 15). For ANM we found weak evidence of reverse causation for PNP and TXNDC15, with only one of the four MR methods obtaining estimates with nominally significant *p*-values (Supplementary Data 16).

**Phenoscanner search.** We used Phenoscanner to look up for GWAS associations of the *cis*-pQTL of the candidate proteins for AAM and ANM with possible confounders of the protein-AAM or ANM association. Our Phenoscanner search revealed associations below the genome-wide suggestive *p*-value threshold of $10^{-5}$ for the majority of the *cis*-pQTL of the MR-prioritized proteins for both AAM and ANM (Supplementary Data 17 and 18). We note a predominance of associations with anthropometric traits, such as weight, height, body mass index (BMI) and body composition measurements. Specifically, among 690 genome-side suggestive associations for 23 *cis*-pQTL of candidate proteins for AAM, 232 associations (around one-third) were with anthropometric traits. Similarly, among 535 genome-suggestive associations for 17 *cis*-pQTL of candidate proteins for ANM, we note 177 associations (around one-third) with anthropometric traits. The *cis*-pQTL with the largest number of genome-wide suggestive associations ($n = 293$) was rs1260326, the MR instrument for GCKR, a protein associated with both AAM and ANM, with a predominance of associations with lipid, glucose metabolism and blood count traits. In summary, for AAM, 8 proteins (ANXA2, ECM1, GCKR, LHB, MST1, NEGR1, SEMA3G, TIE1) showed low evidence of pleiotropy (i.e., < 40% of the total GWAS associations were pleiotropic) while this applied for 8 proteins for ANM (C4A C4B, GCKR, ITPKA, LY9, PLXNA1, PPA1, TNFRSF17, TYMP).

**Two-step network MR and multivariable MR.** In order to explore a mediating effect of BMI in our candidate protein-AAM or candidate protein-ANM associations, we conducted a two-step network MR[25] with BMI as a potential mediator (Fig. 3) (Supplementary Data 19 and 20). For the AAM analysis, we used a large childhood BMI GWAS by Vogelezang et al.[26] while in the

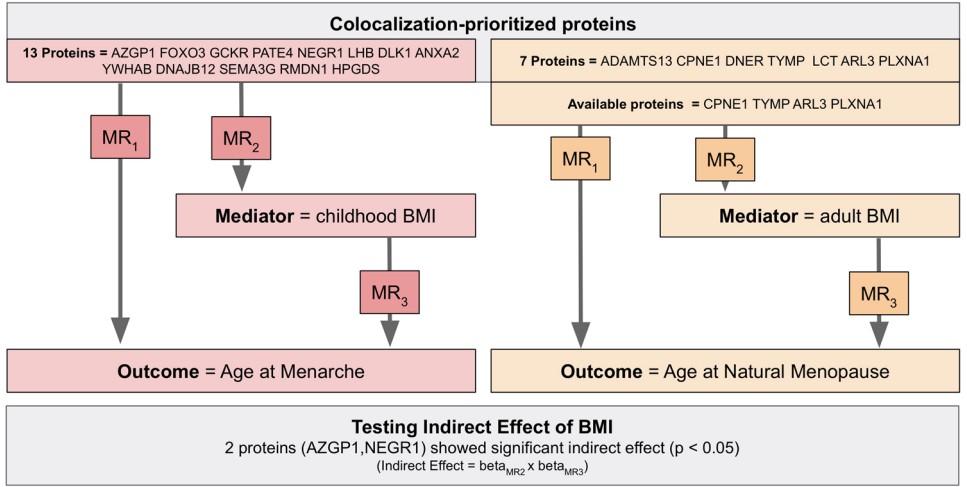

**Fig. 3 Two-step network MR analysis for AAM and ANM.** For the AAM analysis, we used, as mediator, a large childhood BMI GWAS by Vogelezang et al while in the ANM analysis we used, as a mediator, a large adult BMI GWAS by Yengo et al. The analyzes were restricted to proteins prioritized by both MR and colocalization, i.e., 13 proteins for AAM and 7 proteins for ANM. See Supplementary Data 19 and 20 for the result.

ANM analysis we used a large adult BMI GWAS by Yengo et al.[27]. The analyses were restricted to proteins prioritized by both MR and colocalization, i.e. 13 proteins for AAM and 7 proteins for ANM. To conduct the two-step MR, we first step was an MR between protein level and BMI, and the second step was an MR between the BMI and the outcome (AAM or ANM). The indirect effect of the protein on the outcome (ie the effect explained by the BMI) was estimated by multiplying the effect of proteins on BMI and the effect of BMI on outcomes. The standard error and the test of the indirect effect were carried out with the Sobel test. Note that for the ANM analysis, *cis*-pQTL for only 4 proteins were available in the GWAS of adult BMI and we could not find proxies for the 3 other proteins in SNiPA.

We demonstrated that the effect of NEGR1 and AZGP1 proteins on AAM was mediated by pediatric BMI. Respectively, for NEGR1 the indirect effect is 0.118 years (mediated proportion = 67%, $p = 0.0026$) while for AZGP1 the indirect effect is 0.047 years (mediated proportion = 48%, $p = 0.0140$).

In addition, we tested the simultaneous effects of BMI and proteins on AAM or ANM by conducting multivariable MR (MVMR[28]- see Supplementary Data 21 and 22). To do this, for each protein, we used its *cis*-pQTL and SNP-instruments for BMI (pediatric or adult), and retrieved their effects on both exposures and on AAM or ANM. For the ANM MVMR, we used 18 SNPs, 1 *cis*-pQTL for each protein and 17 independent genome-wide significant SNPs for pediatric BMI. For ANM, we used 522 SNPs, 1 *cis*-pQTL for each protein and 521 independent SNPs from the BMI GWAS. Our MVMR analyses for AAM showed that pediatric BMI had significant effects on AAM, after considering the simultaneous effect of each of 13 candidate proteins. Contrarily the MR effect of all 13 proteins was non-significant after accounting for pediatric BMI. In the MVMR analyses for ANM, the effect of adult BMI was consistently not significant, but two proteins (ADAMTS13 and CPNE1) maintained a significant effect on ANM after accounting for adult BMI.

**Protein-altering variant (PAV) assessment.** GWAS-identified *cis*-pQTL may be, or may not be PAVs or in LD with these. Assessing this is important, since PAVs can influence the measurement of candidate proteins by altering the affinity and binding of the molecules used to quantify the proteins level[29]. We assessed if the *cis*-pQTL of the proteins prioritized by colocalization for AAM and ANM were PAVs, or in LD ($r^2 > 0.8$) with

PAVs. None of the *cis*-pQTLs of proteins related to AAM were missense variants, suggesting the absence of protein-altering effects for these proteins (Supplementary Data 23). Contrarily, rs1049564, the *cis*-pQTL for PNP, and the *cis*-pQTL for GCKR (rs35073769), both candidate proteins for ANM, are missense variants, and therefore could be subject to potential binding effects (Supplementary Data 24).

**Pathway and enrichment analyses.** All proteins with positive evidence for colocalization (H4 > 0.8 using colocalization with or without the SuSiE plug-in), or in total 13 proteins for AAM and 7 proteins for ANM, were retained for downstream pathway and enrichment analyses, to further explore the function of the candidate proteins.

Our GeneMANIA analysis showed that the majority of genes encompassing the 13 candidate proteins colocalizing with AAM had physical interaction, co-expression, co-localization or shared a biological pathway (Supplementary Data 25). Using Metascape, we found that 4 genes (*AZGP1, DLK1, GCKR* and *NEGR1*) are linked to diabetes (Supplementary Data 26). *FOXO3* and *YWHAB* are linked to neurodevelopmental disease, while *FOXO3* and *ANXA2* are associated with cardiovascular diseases. *LHB* has been linked to delayed puberty. Finally, 8 out of the 13 genes have been associations with various types of cancer (predominantly breast, prostate and lung cancers). Gene network, enriched ontology, and protein-protein interaction[30] analyses in relation to AAM highlighted the involvement of *DLK1* in notch signaling and of *HPGDS* in prostaglandin metabolic process as well as unsaturated fatty acid biosynthetic process (FDR *P*-value < 0.05) (Fig. 4a). Using FUMA, we were able to demonstrate that the genes of the AAM-related proteins were mainly expressed in the following tissues: adrenal gland, adipose subcutaneous tissue, pituitary and esophagus gastroesophageal junction (Fig. 5a, b, Supplementary Data 27).

Among the genes of the 7 colocalized proteins for ANM, GeneMANIA showed that *TYMP, LCT* and *SLC5A1* (the latter having a physical interaction with *LCT*) play a role in digestion (Fig. 4b, Supplementary Data 28). Also, we found evidence that *TYMP* and *PLXNA1* are linked to neurodevelopmental functions. Similar results were obtained for the two latter genes in the enrichment analysis using Metascape (Supplementary Data 29), which also highlighted the involvement of *CPNE1* and *PLXNA1* in breast cancer. Concerning the tissue expression of genes of the

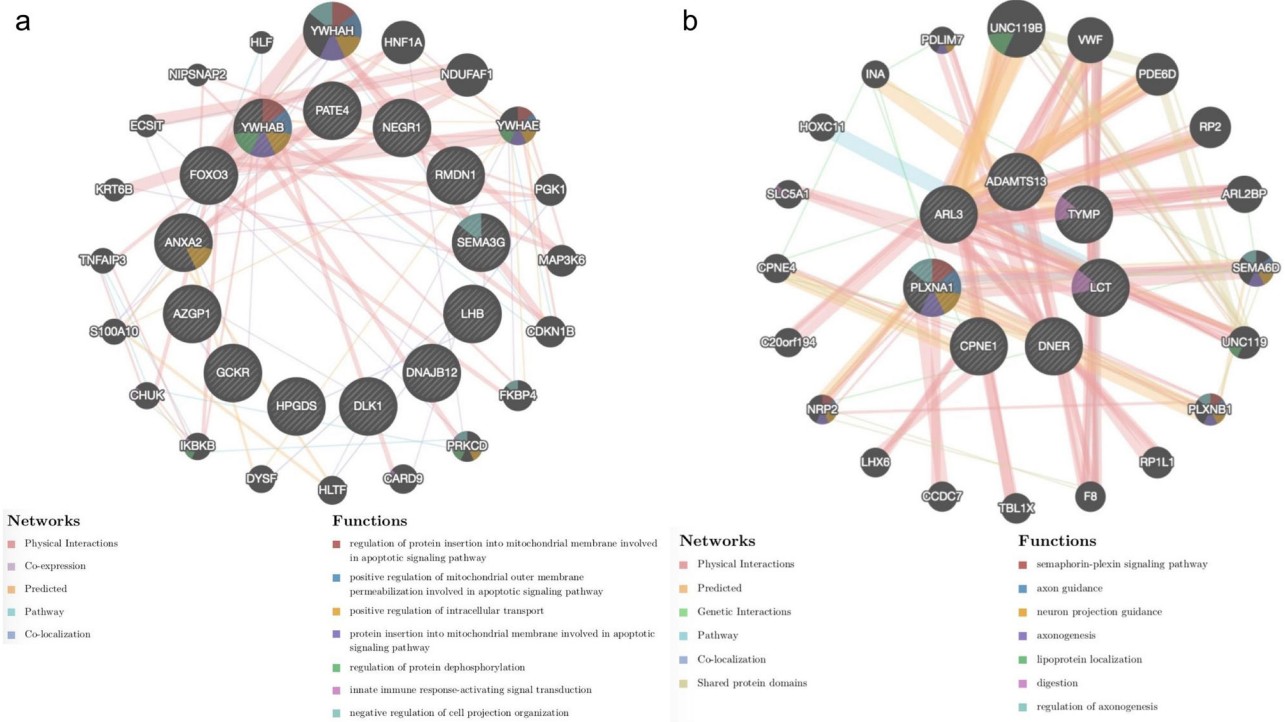

**Fig. 4 Protein-protein interactions and gene network analyses using GeneMANIA for 13 and 7 candidate proteins for AAM and ANM respectively.** The genes of the coloc-prioritized proteins (**a** for AAM and **b** for ANM) are shown as larger inner circles, while genes from the GeneMANIA extension are smaller and appear in the outer circle. Here, the red line represents the Physical Interaction, Purple line represents Co-Expression, Green line represents the Genetic interaction, and blue line represents Pathway and Co-location. (For interpretation of the color references in this figure legend, the reader is referred to the Web version of this article.) The dark gray circle or node size represents the weights value: the larger nodes indicate greater weights. See Supplementary Data 25 and 28 for more information.

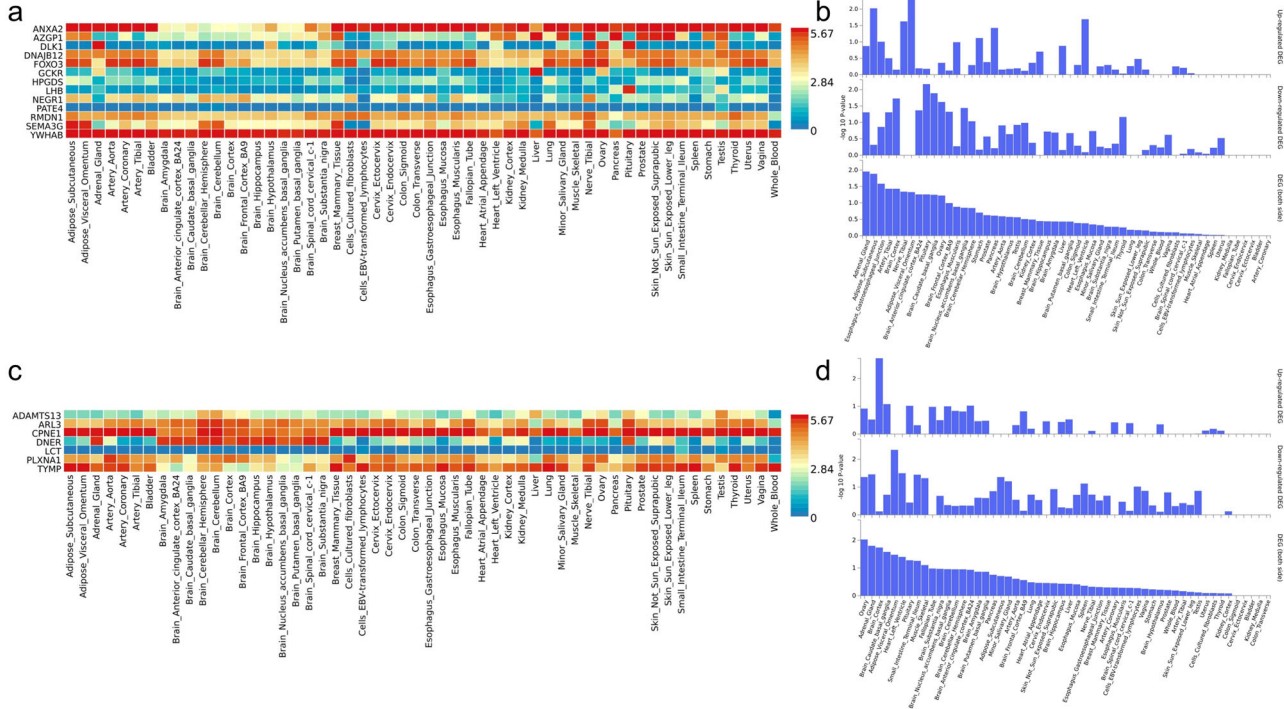

**Fig. 5 Clustered tissue expression heatmaps of the *cis*-pQTL of the candidate proteins for AAM and ANM from FUMA8.** The graph (**a** for AAM and **c** for ANM) depicts normalized expression value (zero mean normalization of log2 transformed expression), where darker red signifies higher relative expression of the gene in each label, compared to a darker blue color in the same label. The graph (**b** for AAM and **d** for ANM) depicts -log10 *p*-values of Differentially Expressed Genes (DEG) sets for each of expression data set. « Up-regulated » denotes over-expression and « Down-regulated » denotes under-expression. See Supplementary Data 27 and 30 for more information.

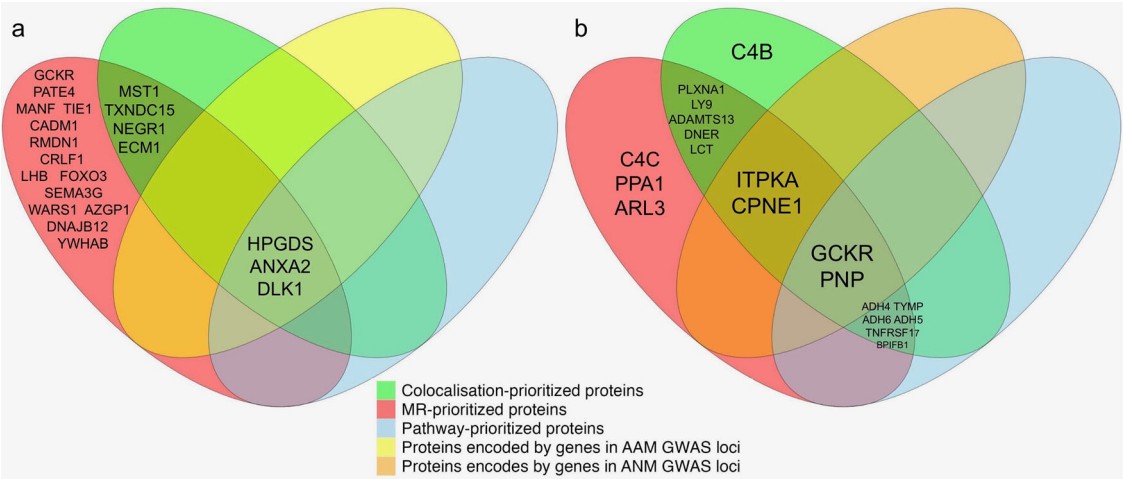

**Fig. 6 Venn diagram summary of results.** Panel (**a**) corresponds to the AAM analysis and (**b**) corresponds to the ANM analysis.

ANM- related proteins, FUMA showed predominant expression in the ovary for *ARL3, CPNE1* and *PLXNA1* (Fig. 5c, d, Supplementary Data 30).

Using OpenTargets (https://www.opentargets.org), we investigated whether any of the prioritized proteins for AAM or ANM were encoded by genes that were known drug targets (Supplementary Data 31 and 32). Among all candidate proteins, genes encoding 4 proteins for ANM (ADH5, PNP, TNFRSF17, TYMP) are targets for at least two existing drugs. Figure 6 shows Venn diagrams summarizing all the results of our study for AAM (Fig. 6a) and ANM (Fig. 6b).

## Discussion
In this MR study, we used *cis*-pQTL as instruments to study putative causal effects of circulating proteins on AAM and ANM. Our MR results were validated using colocalization, and provide evidence for an association between 13 circulating proteins (GCKR, FOXO3, SEMA3G, PATE4, AZGP1, NEGR1, LHB, DLK1, ANXA2, YWHAB, DNAJB12, RMDN1 and HPGDS) and AAM, and between 7 circulating proteins (CPNE1, TYMP, DNER, ADAMTS13, LCT, ARL, and PLXNA1) and ANM. Our reverse MR analysis showed strong evidence of reverse causation for LHB, meaning that AAM causes altered levels of this protein and not the opposite. Using network MR, we showed that the effect of 2 proteins (NEGR1 and AZGP1) on AAM was mediated by BMI, while the *cis*-pQTL of the majority of the proteins demonstrate GWAS associations with anthropometric traits, which could represent confounders of the protein-AAM or protein-ANM associations. Our MVMR analyses also suggest a possible mediating or confounding effect of BMI in the association of candidate proteins and AAM, and the same applied for most candidate proteins for ANM, with the exception of ADAMTS13 and CPNE1.

Despite the relatively small individual effects of a genetically determined one SD change in the candidate proteins on menarche or menopause timing (the largest MR effect being 1,8 years for ANM), these findings are important, since pharmacological inhibition of the entire gene encoding a protein can have magnified effects on both traits.

Our unbiased approach using MR followed by colocalization highlighted 13 candidate proteins with evidence from previous GWAS of associations of their genes with AAM[17]. We identified three proteins (HPGDS, ANXA2 and DLK1) which have previously been associated with AAM in GWAS[16]. Of them, the gene encoding DLK1 (delta like non-canonical Notch ligand 1), is known to be involved in monogenic cases of precocious puberty[31].

Moreover, we were able to cluster the prioritized proteins in biological processes using pathway analysis. For instance, HPGDS (hematopoietic prostaglandin D synthase) has been involved in the metabolism of prostaglandins, and promotes neuroinflammation in Alzheimer's disease[32]. In line with the findings of our gene network analyses, previous observational and MR studies have identified associations between fatty acids and puberty timing[33]. ANXA2 (annexin A2) has also been involved in the regulation of lipoproteins, while its *cis*-pQTL (rs36032232) is in LD with a *cis*-pQTL in the harbor Related Orphan Receptor A (rs3743266) which has been associated with timing of menarche (57).

Our pathway enrichment analysis provided evidence of involvement of candidate proteins for AAM in various cancers. In keeping with these findings, there is previous evidence association of HPGDS with a number of cancer types: colorectal[30], endometrial, ovarian, and thyroid cancer (https://www.proteinatlas.org). *ANXA2* overexpression has been linked to poor outcomes in many cancers, making ANXA2 a candidate biomarker for cancer prognosis[34], and a target for cancer treatment in animal[35] and in vitro experiments[36]. In humans, ANXA2 has been identified as a poor prognostic factor for endometrial cancer[37], estrogen receptor negative (ER-) breast cancer and its subtypes[38], while it promotes endometrial cells' proangiogenic capacity[39]. Our enrichment analysis results support previous findings, showing that DLK1 and HPGDS were involved in acute myeloid leukemia.

In regards to ANM, 2 genes encompassing *cis*-pQTL for candidate proteins (CPNE1 and GCKR) were known GWAS loci for menopausal timing[40,41]. Our pathway and enrichment analyses showed involvement of several of our candidate proteins for ANM in a variety of cancers. In this regard, it is known that longer exposure to reproductive cycling through earlier AAM and later ANM increases the risk of various cancers in women[42]. ANM is determined by the non-renewable ovarian reserve, and oocytes' capacity to maintain genetic integrity and DNA damage response mechanisms appears to play a key role in ANM[18]. In agreement with this, it has been shown that *CPNE1* knockout mice were sensitive to radiation, suggesting that CPNE1.

(copine 1) may be involved in the DNA damage response[43]. In humans, DNER[44] has been associated with breast cancer, TYMP (thymidine phosphorylase) with liver tumours[45], and PLXNA1 (plexin A1) with lung cancer[46].

Among our MR-prioritized proteins for ANM, LCT, the lactase enzyme, also colocalized with ANM. Galactose exerts effects on ovarian function only at extremely high concentrations in rodent models[47]. Lactose and galactose consumption in humans have been linked to a later onset of natural menopause[48].

The *cis*-pQTL (rs48988235) of LCT has been previously associated with improved glucose metabolism in menopausal obese females[49]. Finally, GCKR, the glucokinase regulatory protein, which was associated with both AAM and ANM in our MR and colocalization analyses, exert effects on glucose homeostasis[50], as demonstrated equally by our Phenoscanner search. Nevertheless, the *cis*-pQTL of this protein appears to be the most pleiotropic among the MR instruments of all candidate proteins, and therefore the link of this protein with AAM and ANM should be interpreted with caution.

In order to verify the direction of causality of our MR-prioritized proteins, we conducted reverse MR analyses, with AAM or ANM as exposures and the levels of circulating proteins as outcomes. These analyses confirmed that the direction of the causal association of our forward MR analyses (where proteins were the exposure and AAM or ANM were the outcomes) was right, except for the LHB protein which showed reverse causality with AAM in three MR methods. LHB, the luteinizing hormone (LH) B subunit, is a gonadotropin secreted by the pituitary following pubertal start, which regulates the reproductive function by controlling the production of sex steroid hormones such as estrogen[51]. Therefore, this finding makes sense, since LHB production is the epiphenomenon of pubertal start and not its cause, and can serve as a positive control for our MR study.

A key strength of this study was its combined MR and colocalization approach, which was completed by a pathway and enrichment analysis, highlighting shared biological pathways among the candidate proteins. This triangulation approach can help unravel causal mechanisms by facilitating integration of diverse types of data. Since pathway boundaries tend to be arbitrary, we used two methods and various pathway definitions yet similar clusters were observed.

There are some limitations in our study. In every MR study, intrinsic pleiotropy of SNPs used as instruments could hinder the reliability of MR results[52]; here we employed single-instrument MR, using *cis*-pQTL directly mapping to the genes encoding the proteins, which limits significantly the risk of pleiotropy, but restricts the use of pleiotropy-robust MR methods applicable only in multi-instrument MR. While our Phenoscanner analysis identified associations of the MR instruments of the candidate proteins with BMI and other anthropometric traits, it is not clear if these associations represent pleiotropic pathways, since changes in these traits can be epiphenomena of variations in pubertal or menopausal timing and not the cause of such variations. Nevertheless, the results of our mediation analysis for BMI imply that for some of the identified proteins, anthropometric traits can explain part of the effects of these proteins on AAM or ANM. Sex heterogeneity may bias the estimates of our MR analysis, since the exposures (protein levels) were measured in sex-combined cohorts. When the similarity assumption of age and sex distribution between gene-exposure and gene-outcome associations is violated, sex-specific and sex-combined data may indeed yield a small amount of bias. Despite this, the MR approach could still provide evidence on whether a causal association exists[53]. Additionally, the levels of proteins in the proteomic GWAS were adjusted for sex, mitigating sex heterogeneity bias in the GWAS effect estimates. Data on AAM and ANM are recalled by the participants and subject to misclassification. However, in one longitudinal study, 84% of women whose mean age was 50, recalled their AAM to within one year of the actual date[54]. Non-differential misclassification of AAM and ANM might have biased associations, most likely towards the null. Another limitation is that the GWAS for protein levels have been conducted in adults, and as such, we had to make an assumption that the effect of pQTL is lifelong when assessing the effect of the proteins on AAM, which is an event occurring earlier in life. Moreover,

our approach tested effects of circulating proteins and could not capture potential tissue-specific effects, ie proteins expressed in the pituitary or other relevant tissues. Recently, it has been suggested that erroneous conclusions from MR methods could arise when the direction of the protein binding affinity and the function of the protein are disconnected[55]; while our PAV assessment accounts for binding effects, residual bias due to this phenomenon cannot be completely excluded. Differences in the directions of the MR estimates for some cross-validated proteins may be explained by this phenomenon. We used Bonferroni correction to protect against type 1 error, a conservative method that increases false negatives and, as a result, reduces statistical power in small sample size. However, our MR study had sufficient power to overcome this limitation. Finally, the findings of our MR study are based on GWAS including populations of European descent and cannot be generalized to other ethnic groups. This further underscores the importance of future GWAS on large-scale biobanks including individuals of non-European descent.

In conclusion, our study combined the MR approach with colocalization and *in-silico* annotation to identify and interpret the causal role of circulating proteins with two phenotypes affecting reproductive longevity. Our findings support a potential utility of these proteins as targets for development or repositioning of drugs to manage extreme variations of AAM or ANM, and their related adverse health outcomes. Further validation and investigation of the shared and specific proteomic profiles affecting AAM and ANM may further help unravel pathophysiological pathways encompassing the female reproductive lifespan and targets for treatment.

## Methods

**Research design and methods**. Our methods and findings are reported according to the MR-STROBE checklist (Supplemental Material). No ethics approval was required for this study. Ethics approvals and informed participant consents were obtained for each of the GWAS studies described in Supplementary Data 1.

**GWAS for circulating proteins**. MR instruments for circulating proteins (*cis*-pQTL) were obtained from six proteomic GWAS[7–11,21]. *Cis*-pQTLs were defined as conditionally independent single-nucleotide polymorphisms (SNPs) robustly associated with the proteins, located in 1 Mb of the transcription start site of coding gene, achieving a *P*-value below the significance threshold in each GWAS (generally $\leq 5 \times 10^{-8}$). Circulating proteins in the Sun et al.[7], Emilsson et al.[8], Suhre et al.[9], Yao et al.[11], and Ferkingstad et al.[21] GWAS were measured using the SomaLogic platform, while in the Folkersen et al. GWAS the O-link platform was used[10] (Supplementary Data 1).

**GWAS on AAM and ANM**. To assess the association of *cis*-pQTL with AAM, we retrieved the effects of these *cis*-pQTL on AAM in 370,000 European women from the REPROGEN Consortium GWAS[16] (Supplementary Data 1). We obtained effects of the *cis*-pQTLs on ANM from the largest GWAS meta-analysis available to date equally from the REPROGEN Consortium[18]. This GWAS included a total sample of 200,000 women of European descent from 21 studies. (Supplementary Data 1).

**Statistical analyses**
*Two-sample MR.* Since generally there was a single *cis*-pQTL per protein, we performed single-instrument two-sample MR studies using the Wald ratio to estimate the effect of each candidate circulating protein on AAM and ANM, using the TwoSampleMR R package version 0.5.5[56]. We looked-up the effects of the lead SNPs (*cis*-pQTL) in the six proteomic GWAS[7–11,21] and did the

same in both outcome GWAS. To compute the Wald ratios, SNP-exposure effects were used against SNP-outcome effects to compute a single MR estimate reflecting the effect of each protein linked to a given *cis*-pQTL on AAM, and ANM. Bonferroni correction was used to control for the total number of distinct proteins tested in our MR experiments. The results were presented as MR beta coefficient (β) and 95% confidence intervals (CIs) representing changes in AAM and ANM in years per genetically predicted 1 SD change in circulating protein level.

The first MR assumption (relevance assumption) was met by employing *cis*-pQTL as instruments which have been associated with their respective protein's level at a genome-wide significant level. For *cis*-pQTL that were not present in the AAM or ANM GWAS, SNPs in high LD (defined by a $r^2 > 0.8$ in the 1000 Genomes phase 3 European panel) were identified as proxies using SNiPA (https://snipa.helmholtz-muenchen.de/snipa3/). As another metric of strength of association of the genetic instrument to the exposure we calculated the F-statistic for each *cis*-pQTL, which should be > 10 to consider a SNP as a strong MR instrument. The F-statistic was calculated using the following formula: $F = (R^2/k)/([1 - R^2]/[n - k - 1])$, where $R^2$ is the proportion of the variance of the respective protein level explained by the *cis*-pQTL, k is the number of instruments used in the model (in this case k=1 since there was a single *cis*-pQTL per protein) and n is the GWAS sample size[57]. To compute the proportion of the variance of the respective protein level explained by the *cis*-pQTL ($R^2$) we used the following formula: $R^2 \approx 2\beta^2 f (1- f)$, where β and $f$ denote the effect estimate and the effect allele frequency of the allele on a standardized phenotype respectively[58].

In regards to the second MR assumption (independence assumption), requiring that the instrumental variables do not share common cause with the outcome, we aimed to eliminate sources of confounding affecting our SNP-instruments. Since a potential source of confounding between exposure and outcome is ancestry, we accounted for this by ensuring that all cases and controls in the GWAS sources used in our MR studies are of European ancestry using Eigenstrat[59]. A potential confounder in the association of MR prioritized proteins with AAM and ANM is BMI, since BMI is a known risk factor for altered timing of both AAM and ANM[60–62]. Thus, we undertook a search using the Phenoscanner database[63], to detect GWAS associations of the cis-pQTL of the MR-prioritized proteins with complex traits at a genome-wide suggestive *p*-value threshold ($< 10^{-5}$). Finally, according to the third MR assumption (exclusion restriction assumption), the effects of the genetic instruments (SNPs) on the outcome are mediated solely through the exposure (*cis*-pQTL in this study). Violation of this assumption is known as horizontal pleiotropy. Potential bias due to horizontal pleiotropy is greatly reduced in our study since the instruments in our main MR studies are *cis*-acting SNPs, meaning that they directly map into the genes that encode the proteins[64]. *Cis*-pQTL are considered to have a higher biological prior for a direct and definite influence on the protein compared to *trans*-pQTL. Steiger directionality test and a reverse MR analysis were undertaken for all MR-prioritized proteins, in order to test if AAM or ANM cause the change in the protein level rather than the opposite. Steiger test calculates the variance explained in the exposure (here the protein level) and the outcome (here AAM or ANM) by an MR instrument (here a *cis*-pQTL), and outputs a 'TRUE' or 'FALSE' result, indicating if the direction of the association is true (in this case the instrument explains more of the variance of the exposure than of the outcome), or false (meaning that the instrument explains a larger portion of the variance of the outcome than of the exposure). For the reverse MR analysis, the same GWAS sources were used as in the main MR analyses but the exposure was AAM or ANM and the outcome was the protein level. Instrumental variables for the

two exposures (AAM and ANM) were defined as genome-wide significant (GWAS *p*-value $< 5 \times 10^{-8}$) and LD- independent ($R^2 < 0.001$) SNPs for both exposures, using the "*clump*" function in the *TwoSampleMR* R package. Since these were multi-instrument MRs, sensitivity MR methods testing for pleiotropy (such as the weighted median, the weighted mode and the MR-Egger) were applied in addition to the main inverse-variance weighted MR analysis, using the same R package. Finally, we undertook two-sample network MR[25] and MVMR[28] to identify a mediating or confounding effect of BMI in the significant protein-AAM or protein-ANM MR associations.

**Colocalization analysis**. To assess whether the MR-prioritized proteins for AAM or ANM share a common causal variant in each protein locus, we performed colocalization analyses, as implemented in the *coloc* R package[65]. To estimate the posterior probability of each genomic locus containing a single variant affecting both the protein and the AAM and ANM, we analyzed all SNPs with minor allele frequency > 0.01 within 1 MB of the *cis*-pQTLs. The analyses were undertaken for proteins with MR evidence for association with AAM and ANM using summary-level results from Sun et al.[7], Suhre et al.[9] or Ferkingstad et al.[21]. Due to unavailability of full GWAS for Emilsson et al. we performed our colocalization for MR prioritized proteins from these GWAS using *cis*-pQTL from Ferkingstad et al.[21]. The results of our co-localization analyses were interpreted as follows: each analysis provided posterior probabilities for H0 (no association of the genomic locus with either trait), H1 (association with AAM or ANM but not with the protein level), H2 (association with the protein level but not with AAM or ANM), H3 (association with AAM or ANM and the protein level through two different SNPs), and H4 (association with AAM or ANM and the protein level through one shared SNP). Note that a limitation of colocalization in its traditional form is the assumption of a single shared common causal SNP, however, in reality, genetic loci may contain several causal SNPs. As such, we performed colocalization with the SuSiE (Sum of Single Effects) plug-in in the *coloc* R package[23], which allows to relax the assumption of a single shared causal variant. To do this, an LD matrix was created using the 1000Genomes phase 3 reference. The interpretation of the posterior probabilities using the SuSiE plug-in is similar to the traditional colocalization. For proteins that were prioritized for both AAM and ANM, we tested simultaneous colocalization of the three traits, using the *HyPrColoc* package in R[24]. The results of this analysis present posterior probabilities explained by causal SNPs for multiple traits to be causal for all traits simultaneously. When no simultaneous colocalization of the traits is detected, the posterior probabilities appear as "NA".

**Pathway and enrichment analysis, gene network analysis, gene expression analysis and protein–protein interaction[30]**. To build a gene network to study the functions of prioritized proteins for AAM and ANM from both MR and colocalization, and investigate protein-protein interactions, we used the Gene Multiple Association Network Integration Algorithm (GeneMANIA) tool[66,67] (http://www.genemania.org/). This tool finds a small set of genes/proteins that are most likely to share a function with selected genes/proteins based on their interactions. It attributes continuous weights varying between 0 and 1 which indicate the strength of co-regulation between the genes.

For each given protein, pathway and process enrichment analysis was carried out for their corresponding gene using the following gene ontology resources: KEGG Pathway, GO Biological Processes, Reactome Gene Sets, Canonical Pathways, CORUM, TRRUST, DisGeNET, PaGenBase, Transcription Factor Targets, WikiPathways, PANTHER Pathway using Metascape

(https://metascape.org/). Notably, *P*-values were calculated based on the accumulative hypergeometric distribution, and q-values were calculated using the Benjamini-Hochberg procedure to account for multiple testing.

We performed enrichment analysis using FUMA8 (https://fuma.ctglab.nl), which prioritized genes in biological pathways and functional categories using the hypergeometric test against gene sets obtained from MsigDB21 and WikiPathways[68]. A Benjamini–Hochberg adjusted *P*-value ≤ 0.05 was performed to account for multiple testing per data source of tested gene sets (e.g., canonical pathways, GO biological processes, hallmark genes). Finally, to determine if the *cis*-pQTL of the prioritized proteins for AAM and ANM had evidence of being expression quantative trait loci (eQTL), we created a gene expression heatmap using Genotype-Tissue Expression (GTEx) v8 implemented in FUMA8[69]. Finally, we identified if the genes encoding the candidate proteins are targets for existing drugs in the OpenTargets repository (https://www.opentargets.org).

**Evaluation for protein-altering variants**. For *cis*-pQTL of our colocalization-prioritized proteins measured on SomaLogic, we assessed the possibility of potential aptamer-binding effects, where the presence of protein altering variants (PAV) may affect protein measurements. Since PAV can influence the binding of a protein to an antibody in enzyme-linked immunosorbent assays used for protein measurements, this information is important for future validation studies of the candidate proteins[29].

**Sample size and power analysis**. A major limitation of MR is insufficient statistical power. Statistical power in this MR study is a function of the variance of the protein levels explained by the protein-increasing allele of the *cis*-pQLT, and the sample size of the AAM and ANM. We used the method of Brion et al. [23] to calculate power (https://shiny.cnsgenomics.com/mRnd/).

**Reporting summary**. Further information on research design is available in the Nature Portfolio Reporting Summary linked to this article.

## Data availability
Source data underlying Figs. 2 and 5 are provided in Supplementary Data 4, 9, 27 and 30. All GWAS used are publicly available and listed in Supplementary Data 1, along with the links and relevant accession numbers.

## Code availability
R codes used to generate the results of the MR and colocalization analyses are available at the following link: https://github.com/jumentib/Nat_Com_MR/ or on Zenodo[70].

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

## Acknowledgements

We thank all participants in the GWAS consortia, data from which were used in this study. K.K.O. and J.R.B.P. are supported by the Medical Research Council (Unit programme: MC_UU_00006/2). D.M. is a Fonds de Recherche du Quebec-Santé (FRQS) Junior 1 Scholar and has received a Career Development Award from ENRICH (Empowering Next-Generation Researchers in Perinatal and Child Health).

## Author contributions

D.M. conceived the study and supervised the analyses. N.Y., B.J., and M.Y. drafted the manuscript and performed the analyses. All authors contributed in study design, reviewing and writing the manuscript. All authors critically reviewed and approved the final version of the manuscript.

## Competing interests

The authors declared no competing interests.
