## [Peer Review File · Communications Biology]

Reviewers' comments:

Reviewer #1 (Remarks to the Author):

The authors outline the unmet need for predictive biomarkers of extreme female reproductive ageing and highlight the role of circulating proteins as a valuable source for biomarker discovery and drug target characterization.

Nonetheless, they highlight limitations of serum proteome measurement including cost, sample processing and bias. Is the argument that genetic variants associated with circulating proteins might form better predictors or be more useful for guiding drug target evaluation? There seems to be a disconnect between the two penultimate paragraphs and more could be said about the use of pQTLs and MR in this context.

What was the reasoning for using 6 proteomic GWAS to identify pQTLs and how were the 6 studies integrated? What if the same pQTLs were identified in multiple studies, where were the estimates obtained from or how were they derived?

"The 22 proteins had all an F-statistic > 10" should read "the 22 pQTLs..."
As well as F-statistic, it would be useful to report on % variance explained for the pQTLs.

How was the cross-validation done? When describing replication using cis-pQTLs from different studies, were these always different cis-pQTLs and were they are the same locus?

Why were only 3 of the proteomic GWAS used for the colocalization analysis?

Describe briefly in the Results section what is meant by H4

How do you know that the attenuation of the effect with adjustment for BMI in the MVMR represents mediation rather than confounding or pleiotropy? In the Methods section you state that BMI may be a confounder in the association. In relation to ANM, you imply effect modification by adult BMI – how was this determined based on MVMR?

Could multiple trait colocalization be done to investigate further the GCK9 locus in relation to both ANM and AAM?

What are the implications of finding potential binding effects from the PAV assessment?

As the authors indicate, the colocalization approach used in the paper assumes one shared causal variant. Did the author consider the use of an approach which allows for multiple causal variants (10.1371/journal.pgen.1009440)?

In the Results section, it would be useful to give some quantification of the causal effects estimated. In the Discussion the authors state that the effects are relatively small but this is not evident in the Results.

In the Discussion the authors mention that it has been suggested that erroneous conclusions can arise when the direction of the protein binding affinity and function of the protein are disconnection – could the authors provide a reference for this?

As well as the limitations regarding the sex-combined and ancestry-specific nature of the protein

GWAS, another limitation is that the GWAS have been solely/primarily conducted in adults. You are therefore making an assumption regarding lifelong pQTLs when assessing the effect of proteins on events occurring earlier in life (i.e. AAM)

Reviewer #2 (Remarks to the Author):

Summary:

The authors investigate the causal effect of proteins on AAM and ANM in this paper. First, they attain 6 proteomic GWAS (summary statistics for association between SNPs and protein level), which constitute the beta_XG (instrumental variables). They also obtain a GWAS for each of AAM and ANM (summary statistics for association between SNPs and AAM/ANM) which constitute the beta_YG. They then run a two-sample Mendelian randomization for each for each protein (to infer a beta_XY).

AAM: 1271 proteins are tested (1265 cis-pQTLs and 6 LD proxies). With a Bonferroni correction cutoff of $3.9e-5$, 22 proteins are identified as potentially causal.

ANM: 1349 proteins are tested. With a Bonferroni correction cutoff of $3.7e-5$, 18 proteins are identified as potentially causal.

Next was a colocalization analysis with the aim of determining if there was simply LD with another SNP.

AAM: 18 of the 22 proteins had publicly available GWAS for variants surrounding that cis-pQTL. Colocalization analysis found 4 proteins had highest probability of hypothesis 4 (H4: the cis-pQTL and the GWAS hit for AAM are the same variant) and 2 proteins had highest probability of hypotheses 3 (H3: the cis-pQTL and GWAS hit for AAM are in LD).

ANM: All 18 of the proteins had publicly available GWAS for variants surrounding the cis-pQTL. Colocalization analysis found 4 proteins had highest probability of hypothesis 4 (H4: the cis-pQTL and the GWAS hit for ANM are the same variant) and 12 proteins had highest probability of hypotheses 3 (H3: the cis-pQTL and GWAS hit for ANM are in LD).

The second assumption of MR was tested with an MVMR adjusting for BMI. That is, The 6 proteins from AAM colocalization and the 16 proteins from ANM colocalization were tested with MVMR.

Adult BMI (for ANM) and childhood BMI (for AAM) were used for this analysis.

Next was a pathway and enrichment analysis, a test as to whether the cis-pQTLs for the AAM/ANM proteins were also eQTLs, and an evaluation for protein-altering variants which could affect protein measurements.

Finally, a power calculation was done.

Recommendations:

Scientific:

1. Please include a p-value q-q plot (actual vs theoretical p-value distribution) for each of AAM and ANM.
2. Please include the allele frequencies in Supplementary table 3B.
3. Please clarify how the multivariable MR was done. Multivariable MR requires more genetic variants than risk factors (PMC4325677). It then seems that if there is exactly one variant per protein, it would not be possible to also include BMI in the analysis. How did this work?

$$\text{Beta}_Y|G = \text{beta_protein}|G * \text{beta}_Y|\text{protein} + \text{beta_BMI}|G * \text{beta}_Y|\text{BMI}$$

Where $Y = \text{AAM or ANM}$, solving for $\text{beta}_Y|\text{protein}$ and $\text{beta}_Y|\text{BMI}$.

4. The variance explained was set to 4.6% via the power calculation. Citation 16 does indeed have such a high variance explained as a median for all the cis-pQTLs. Yet for those identified in this analysis (Supplementary tables 3), 9/26 and 6/20 have a R^2 above 4.6 %, so the median is lower here. I wonder how this power calculation would change if the median variance explained were more reflective of the cis-pQTLs that surfaced, as well as if the betas tested (Supp table 9) were closer to the median of the betas for the cis-pQTLs selected in Supp. Table 2a.

Clarity within paper:

5. Please number tables of proteins such as in Supplementary table 3B (1-26 and 1-20, respectively), 4A-4B, so the reader does not have to count.

6. The number of proteins still "in the running" at different steps seems inconsistent throughout this paper. It would be helpful to include a flow chart of the number of proteins associated with each phenotype and the methods used to subsequently pare down the list of proteins. I see this partly in Figure 1, but it doesn't seem to line up with tables and text. For example, I expected to see 22 and 18 proteins after the two-sample MR, but there were 26 and 20 in Supplementary table 3B. Also, the text says there was colocalization for 4 proteins for ANM (CPNE1, TYMP, DNER, ADAMTS13) but Figure 1 also shows a 5th protein (LY9). Then Figure 2A caption claims to plot the 22 proteins from AAM but only shows 21 proteins. The text in line 212 refers to 17 proteins but points the reader to Figure 4B, which has 15 proteins; this is after lines 147-151 refer to $4 + 12 = 16$ proteins that colocalize with ANM. Perhaps there needs to be some mapping between the figure and text and tables so that a checkpoint in the figure can also be referred to in the text and the table (for example; any way that makes sense is fine). Otherwise the reader is cross checking and cannot follow. Please make all of this super clear to the reader. Captions can also be expanded in the supplement to make sure the reader can follow how the proteins listed came to be there.

7. Six proteins for AAM MVMR were tested; only 5 are in Supp. Table 5a. Fifteen proteins are listed for ANM MVMR in Supp. Table 5b. Is one left out, or were summary stats not available for the last one? Please specify. Lines 153-154 do not say how many were tested.

8. Please consider sorting Supp Table 5b by p value for the protein and denoting the cutoff as you did for the other table – that was helpful. It would also be helpful to have a side by side comparison of the univariate vs multivariate MR analyses (either a plot or a table that shows both). Further, some sort of shading for every other line or every other protein would make this table more readable.

Minor/copy editing items:

1. There may be a typo in Supplementary 3B. Both tables refer to AAM, and both tables are numbered 3B.

2. Supp Table 5A: beta and pleiotropy are misspelled.

3. Lines 427-429, please improve this sentence – something may be missing here, possibly a preposition.

4. There may be a typo on line 434 (the number 10).

5. Where color denotes meaning (supp table 4B for example), please include the color in the key (e.g. highlight the word orange in the caption in orange, or add an orange box).

Reviewer #3 (Remarks to the Author):

Yazdanpanah et al has delivered a well written manuscript where they have identified circulating proteins as biomarkers for age at menarche and age at menopause. I have some comments below:

*How was power calculations done - should be different based on variance explained by instrument in each analysis.

*No discussion on the possibility of reverse causality particularly in the case for AAM since the protein levels would have been measured in adult. How can the authors be sure the causal relationship is not the other way around?

*I would have liked to see a look-up of the SNPs used as instruments. Are they associated with other phenotypes known to influence AAM/ANM? If so this could violate the no pleiotropy assumption, which is difficult to access currently as no sensitivity analysis could be preformed.

Please find below our answers to the Reviewer's comments in blue.

Reviewers' comments:

Reviewer #1 (Remarks to the Author):

The authors outline the unmet need for predictive biomarkers of extreme female reproductive ageing and highlight the role of circulating proteins as a valuable source for biomarker discovery and drug target characterization. Nonetheless, they highlight limitations of serum proteome measurement including cost, sample processing and bias. Is the argument that genetic variants associated with circulating proteins might form better predictors or be more useful for guiding drug target evaluation? There seems to be a disconnect between the two penultimate paragraphs and more could be said about the use of pQTLs and MR in this context.

RESPONSE: We thank the reviewer for this suggestion. We have rephrased the two last paragraphs of the introduction as follows (lines 24 – 42):

“Circulating proteins in the blood have been a valuable source for biomarker discovery and drug target characterisation for various diseases ⁷. However, the serum proteome can be highly sensitive to sample processing. Moreover, serum proteome measurement is costly and prone to potential bias due to differences in the sample groups. Case-control studies comparing proteomic profiles between individuals affected with diseases versus controls can be subject to reverse causation, but if our goal is prevention, it is necessary to detect changes in proteomic profiles prior to disease onset. Due to the above limitations, it is complicated to comprehensively characterize and compare the serum/plasma proteome in case-control studies ⁸. In addition, better evidence is needed to understand if circulating proteins play a causal role in female reproductive aging.

In the recent years, large-scale proteomic GWAS have identified thousands of variants controlling levels of circulating proteins ¹⁶⁻²⁰. Also, AAM and ANM are highly heritable traits with estimates of heritability for both AAM and ANM varying between 50% and 80% ⁹⁻¹². These estimates have been confirmed by recent large-scale genome-wide association studies (GWAS) on AAM and ANM, leveraging data from millions of women ^{13,14,15}. These GWAS provide a valuable opportunity to utilize Mendelian randomization (MR), a method allowing to test causal associations between biomarkers (exposures, such as circulating proteins) and outcomes, such as AAM and ANM”

What was the reasoning for using 6 proteomic GWAS to identify pQTLs and how were the 6 studies integrated? What if the same pQTLs were identified in multiple studies, where were the estimates obtained from or how were they derived?

RESPONSE: We used 6 proteomic GWAS with distinct populations and non-overlapping with the samples of the outcome GWASes.

The rationale was to maximize discovery of MR associations, by increasing the number of tested proteins among the 6 GWAS. For proteins tested in more than one GWAS, we performed separately MR using *cis*-pQTL for each GWAS (these *cis*-pQTL could be or could not be similar across GWAS). As such, we considered the presence of significant MR associations in multiple GWAS as replication, providing additional support to our MR findings.

*“The 22 proteins had all an F-statistic > 10” should read “the 22 pQTLs...”
As well as F-statistic, it would be useful to report on % variance explained for the pQTLs.*

RESPONSE: We corrected the sentence referring to the F-statistic as requested by the reviewer. The information on the R^2 (variance explained) of the *cis*-pQTL of the MR-prioritized proteins for AAM and ANM appears in Supplemental Tables 3A, and 3B, and the following sentences were added (lines 75 – 79) : “The *cis*-pQTL of the 22 proteins had all an F statistic > 10, ranging from 35.12 (NEGR1) to 25208.6 (TXNDC15), explaining from 0.1% (FOXO3) to 77.1% (MST1) of the variance of the protein levels” and “All 18 *cis*-pQTL of the candidate proteins had F statistics > 10 ranging from 47.4 (TYMP) to 1273.3 (CPNE1), and explained between 0.2% (ITPKA) and 28.5% (CPNE1) of the variance in the protein levels (Supplementary Table 3B)”.

How was the cross-validation done? When describing replication using cis-pQTLs from different studies, were these always different cis-pQTLs and were they at the same locus?

RESPONSE: By definition, all *cis*-pQTLs in all 6 proteomic GWAS were harboring the same respective protein locus (see Methods section, lines 343 – 345): “*Cis*-pQTLs were defined as conditionally independent single-nucleotide

polymorphisms (SNPs) robustly associated with the proteins, located in 1Mb of the transcription start site of coding gene”.

To respond to the reviewer's latter question, we added the following phrases (lines 87 – 89 ; lines 119 – 124): “The *cis*-pQTL associated with the cross-validated proteins differed between the two proteomic GWAS-sources for all of the 6 aforementioned proteins” and “Furthermore, we cross-validated the findings for 4 proteins (LCT, PPA1, ARL3 and PLXNA1) in at least two GWAS studies except CPNE1 which replicated across 3 proteomic GWAS studies (Supplementary Table 2B). The *cis*-pQTL for LCT and PPA1 were identical in the two proteomic GWAS sources, but for the two other cross-validated proteins (PPA1 and CPNE1), the *cis*-pQTL differed.” Also, we highlighted in grey the cross-validated proteins in Supplementary Tables 2A and 2B.

Why were only 3 of the proteomic GWAS used for the colocalization analysis?

RESPONSE: We could perform colocalization only for those proteomic GWAS sources that provided full genome-wide summary statistics. This is described in the following sentence in the results section (lines 94 – 96): “For 18 of the 22 MR-prioritized proteins, genome-wide summary-level GWAS results were publicly available in the Sun et al.¹⁶, Suhre et al.¹⁸, or Ferkingstad et al.²³ GWAS allowing colocalization analyses”.

Describe briefly in the Results section what is meant by H4

RESPONSE: We added some additional explanations in the results section. The phrase now reads as follows (lines 90 – 94): “To assess if our MR results were driven by confounding due to linkage disequilibrium (LD), we tested whether the MR-prioritized proteins shared a single causal variant with AAM using colocalization, where a posterior probability 4 (H4) indicates presence of a single causal SNP for the protein and AAM, while a posterior probability 3 (H3) indicated the presence of two causal variants (one for the protein and the other for AAM) in LD”.

How do you know that the attenuation of the effect with adjustment for BMI in the MVMR represents mediation rather than confounding or pleiotropy? In the Methods section you state that BMI may be a confounder in the association. In relation to ANM, you imply effect modification by adult BMI – how was this determined based on MVMR?

RESPONSE: The multivariable MR analysis was removed from the manuscript, the reason being that, in order to perform this analysis, the number of tested exposures should be smaller than the number of SNP-instruments used for the main exposure of interest—here the protein levels. Since we have undertaken single instrument MR using one *cis*-pQTL per protein, and we would require at least three SNPs as MR-instruments to test two exposures (ie the protein level and BMI), the multivariable MR results were not valid and were removed. Instead, we have added the results of a look-up in Phenoscanner for associations of the *cis*-pQTL of the candidate proteins with complex traits (Supplementary tables S6A and S6B). We observed an increased number of associations with anthropometric traits, including BMI, and body composition measurements.

The interpretation of these GWAS associations as being due to mediation, confounding or pleiotropy is difficult, since these changes could be epiphenomena of altered timing of puberty or menopause and not their causes. We have added a phrase in the discussion on this (lines 296 – 300): “While our Phenoscanner analysis identified associations of the MR instruments of the candidate proteins with BMI and other anthropometric traits, it is not clear if these associations represent pleiotropic pathways, since changes in these traits can be epiphenomena of variations in pubertal or menopausal timing and not the cause of such variations”.

Could multiple trait colocalization be done to investigate further the GCK9 locus in relation to both ANM and AAM?

RESPONSE: We used the hypercoloc R package to test simultaneous colocalization between the GCKR, AAM and ANM. The results are now included in the results section (lines 139 – 142): “By using the R package “hypercoloc” we did not find evidence supporting that GCKR colocalized simultaneously with both AAM and ANM.”

What are the implications of finding potential binding effects from the PAV assessment?

RESPONSE: For all *cis*-pQTL of our MR-prioritized proteins, we assessed the possibility of potential aptamer-binding effects, in which the presence of protein-altering variants (PAVs) may affect protein measurements. Since PAVs can influence the binding of a protein to an antibody in ELISAs used for protein measurements, this information is important for future validation studies of the candidate proteins. As mentioned in our results section (lines 180 – 182), “Assessing this is important, since PAVs can influence the measurement of candidate proteins by altering the affinity and binding of the molecules used to quantify the proteins’ level”. Also, in our methods section, we mention that (lines 469 – 471): “Since PAV can influence the binding of a protein to an antibody in enzyme-linked immunosorbent assays used for protein measurements, this information is important for future validation studies of the candidate proteins”.

As the authors indicate, the colocalization approach used in the paper assumes one shared causal variant. Did the author consider the use of an approach which allows for multiple causal variants (10.1371/journal.pgen.1009440)?

RESPONSE: We thank the reviewer for this suggestion. By testing coloc with SuSiE, we found that additional proteins colocalized when considering more than one causal variants in the 1 Mb region around the cis-PQTL. We updated Supplementary Tables 4A and 4AB with the results of this analysis. We have included all proteins that colocalized with SuSiE in the follow-up analyses. The text of the manuscript and the Figures were modified accordingly to reflect the changes in the number of colocalized proteins (see Figure 1).

In the Results section, it would be useful to give some quantification of the causal effects estimated. In the Discussion the authors state that the effects are relatively small but this is not evident in the Results.

RESPONSE: We added the following two sentences in the Results section to describe the magnitude of the MR effects (lines 77 – 79 ; lines 115 – 117):

“The 22 proteins presented MR beta coefficients (β) on AAM ranging from -0.45 (FOXO3) to 0.46 (MANF) years per 1 SD increase in protein levels (Fig. 2A, Supplementary Table 3A).”

“...we identified 18 proteins significantly associated with ANM, with MR beta coefficients (β) ranging from -0.80 (PNP) to 1.80 years (ITPKA) per 1 SD increase in protein levels (Fig. 2B)”

In the Discussion the authors mention that it has been suggested that erroneous conclusions can arise when the direction of the protein binding affinity and function of the protein are disconnection – could the authors provide a reference for this?

RESPONSE: We have added the study by Jarmoskaite et al, ELife 2020, in our references.

As well as the limitations regarding the sex-combined and ancestry-specific nature of the protein GWAS, another limitation is that the GWAS have been solely/primarily conducted in adults. You are therefore making an assumption regarding lifelong pQTLs when assessing the effect of proteins on events occurring earlier in life (i.e. AAM).

RESPONSE: Indeed, this assumption is made in the AAM MR, where the outcome was measured in a younger population than that of the protein measurement. This was added in the limitations section of the discussion (lines 310 – 312): “Another limitation is that the GWAS for protein levels have been conducted in adults, and as such, we had to make an assumption that the effect of pQTL is lifelong when assessing the effect of the proteins on AAM, which is an event occurring earlier in life”.

Reviewer #2 (Remarks to the Author):

Recommendations:

Scientific:

1. Please include a p-value qq plot (actual vs theoretical p-value distribution) for each of AAM and ANM.

RESPONSE: We are providing for the reviewer’s information the qqplots of the p-value distribution for both AAM (left panel) and ANM (right panel) MR studies. Since we did not observe any inflation suggestive of false positive p-values, we opted not to add these plots in the paper.

2. Please include the allele frequencies in Supplementary table 3B.

RESPONSE: We have added the effect alleles and their frequencies in Supplementary Tables 3A and 3B.

3. Please clarify how the multivariable MR was done. Multivariable MR requires more genetic variants than risk factors (PMC4325677). It then seems that if there is exactly one variant per protein, it would not be possible to also include BMI in the analysis. How did this work?

$\text{Beta}_Y|G = \beta_{\text{protein}}|G * \beta_{\text{Y}}|_{\text{protein}} + \beta_{\text{BMI}}|G * \beta_{\text{Y}}|_{\text{BMI}}$
 Where Y = AAM or ANM, solving for $\beta_{\text{Y}}|_{\text{protein}}$ and $\beta_{\text{Y}}|_{\text{BMI}}$.

RESPONSE: We thank the reviewer for identifying this error. Indeed, in order to perform the multivariable MR analysis, the number of tested exposures should be smaller than the number of SNP-instruments used for the main exposure of interest—here the protein levels. Since we have undertaken single instrument MR using one cis-pQTL per protein, and we would require at least three SNPs as MR-instruments to test two exposures (ie the protein level and BMI), the multivariable MR results were not valid and were removed.

4. The variance explained was set to 4.6% via the power calculation. Citation 16 does indeed have such a high variance explained as a median for all the cis-pQTLs. Yet for those identified in this analysis (Supplementary tables 3), 9/26 and 6/20 have a R^2 above 4.6 %, so the median is lower here. I wonder how this power calculation would change if the median variance explained were more reflective of the cis-pQTLs that surfaced, as well as if the betas tested (Supp table 9) were closer to the median of the betas for the cis-pQTLs selected in Supp. Table 2a.

RESPONSE: In order to calculate precisely the power of our MR study, we computed the power for each of MR-prioritized proteins based on its R^2 . The results appear in Supplementary Table 11. We rephrased this part of the results section as follows (lines 63 – 69):

“The results of the MR analysis, expressing changes in AAM or ANM in years per 1 standard deviation (SD) increase in the levels of a circulating protein, are presented in Fig. 2. Our power analysis showed that at an $\alpha = 0.05/1,300 = 3.8 \times 10^{-5}$ (after Bonferroni correction), our power to detect an effect as small as 0.13 years in AAM or of 0.47 years in ANM per SD change in a specific protein level was 100% for the majority of the MR-prioritized proteins, based on the variance explained of each protein, and a reported SD for AAM of 1.3 years and for ANM of 4 years (Supplementary Table 11).”

Clarity within paper:

5. Please number tables of proteins such as in Supplementary table 3B (1-26 and 1-20, respectively), 4A-4B, so the reader does not have to count.

RESPONSE: Since not all proteins appearing in Tables S3A and S3B are included in Tables 4A and 4B (colocalization), we ordered the proteins in all 4 tables in alphabetical order to facilitate the reader.

6. The number of proteins still “in the running” at different steps seems inconsistent throughout this paper. It would be helpful to include a flow chart of the number of proteins associated with each phenotype and the methods used to subsequently pare down the list of proteins. I see this partly in Figure 1, but it doesn’t seem to line up with tables and text. For example, I expected to see 22 and 18 proteins after the two-sample MR, but there were 26 and 20 in

Supplementary table 3B. Also, the text says there was colocalization for 4 proteins for ANM (CPNE1, TYMP, DNER, ADAMTS13) but Figure 1 also shows a 5th protein (LY9). Then Figure 2A caption claims to plot the 22 proteins from AAM but only shows 21 proteins. The text in line 212 refers to 17 proteins but points the reader to Figure 4B, which has 15 proteins; this is after lines 147-151 refer to $4 + 12 = 16$ proteins that colocalize with ANM. Perhaps there needs to be some mapping between the figure and text and tables so that a checkpoint in the figure can also be referred to in the text and the table (for example; any way that makes sense is fine). Otherwise the reader is cross checking and cannot follow. Please make all of this super clear to the reader. Captions can also be expanded in the supplement to make sure the reader can follow how the proteins listed came to be there.

RESPONSE: Given that the number of proteins which colocalized has changed with the addition of the coloc using SuSiE, the Figure 1 was entirely redone.

In regards to the reviewer's comment on the number of proteins in Supplemental Tables S3A and S3B: Among the 27 AAM proteins in Tables S2B, there are 22 unique proteins (since 5 proteins were identified in more than one MRs), which explains the difference depicted by the reviewer. Similarly, among the 24 proteins in Tables S2B and S3B, there are 18 unique proteins (since 5 proteins were identified in more than one MRs). The correction in the number of colocalized proteins was made (7 proteins in total for AAM, with the addition of GCKR). The 22nd protein for AAM (ECM1) was added in Figure 2A. We added the 16th protein (C4B) in the heatmap demonstrated in Figure 4B. The text in line 212 was changed (16 instead of 17 proteins).

7. Six proteins for AAM MVMR were tested; only 5 are in Supp. Table 5a. Fifteen proteins are listed for ANM MVMR in Supp. Table 5b. Is one left out, or were summary stats not available for the last one? Please specify. Lines 153-154 do not say how many were tested.

RESPONSE: Please see above answer to comment 3 of the reviewer. The MVMR analysis was invalid and was removed.

8. Please consider sorting Supp Table 5b by p value for the protein and denoting the cutoff as you did for the other table – that was helpful. It would also be helpful to have a side by side comparison of the univariate vs multivariate MR analyses (either a plot or a table that shows both). Further, some sort of shading for every other line or every other protein would make this table more readable.

RESPONSE: We have sorted Supplemental Table 5b by CADD PHRED. As mentioned above, we removed the MVMR analysis.

Minor/copy editing items:

1. There may be a typo in Supplementary 3B. Both tables refer to AAM, and both tables are numbered 3B.
2. Supp Table 5A: beta and pleiotropy are misspelled.
3. Lines 427-429, please improve this sentence – something may be missing here, possibly a preposition.
4. There may be a typo on line 434 (the number 10).
5. Where color denotes meaning (supp table 4B for example), please include the color in the key (e.g. highlight the word orange in the caption in orange, or add an orange box).

RESPONSE: In the present version, all the editing comments of the reviewer were addressed.

Reviewer #3 (Remarks to the Author):

Yazdanpanah et al has delivered a well written manuscript where they have identified circulating proteins as biomarkers for age at menarche and age at menopause. I have some comments below:

*How was power calculations done - should be different based on variance explained by instrument in each analysis.

RESPONSE: Please see above answer to comment 4 of reviewer 2. The power analysis was redone entirely taking into consideration the R^2 of each protein by its instrument.

*No discussion on the possibility of reverse causality particularly in the case for AAM since the protein levels would have been measured in adult. How can the authors be sure the causal relationship is not the other way around?

RESPONSE: To examine this possibility, we have performed a reverse MR for the MR-prioritized proteins (ie AAM or ANM were the exposures, and protein levels were the outcomes) using the IVW approach, and 3 other pleiotropy-

robust methods (Egger, weighted median and weighted mode). Taken together, the results of these MR analyses failed to consistently show any causal association between AAM or ANM and the candidate protein levels, with the exception of the LHB, where 3 out of the 4 MR methods showed an association, with p-values approaching the Bonferroni-corrected p-value threshold. This analysis appears now in the results section and in Supplementary Tables 5A and 5B.

*I would have liked to see a look-up of the SNPs used as instruments. Are they associated with other phenotypes known to influence AAM/ANM? If so this could violate the no pleiotropy assumption, which is difficult to access currently as no sensitivity analysis could be performed.

RESPONSE: The risk of pleiotropy using as MR instruments cis-pQTL mapping in the gene encoding the circulating proteins is low. Nevertheless, we have done a look-up in Phenoscanner for the SNP-instruments of the MR-prioritized proteins for AAM and ANM. We note a predominance of associations with anthropometric traits, such as height, weight, BMI and body composition measurements. The results appear in Supplemental Tables 6A (for AAM) and 6B (for ANM), and in the results section (lines 167 – 177): “Phenoscanner search”:

“Our Phenoscanner search revealed associations below the genome-wide suggestive p-value threshold of 10^{-5} for the majority of the cis-pQTL of the MR-prioritized proteins for both AAM and ANM (Supplementary Tables 6A and 6B). We note a predominance of associations with anthropometric traits, such as weight, height, BMI and body composition measurements. Specifically, among 690 genome-wide suggestive associations for 23 cis-pQTL of candidate proteins for AAM, 232 associations (around one third) were with anthropometric traits. Similarly, among 535 genome-wide suggestive associations for 17 cis-pQTL of candidate proteins for ANM, we note 177 associations (around one third) with anthropometric traits. The cis-pQTL with the largest number of genome-wide suggestive associations ($n=293$) was rs1260326, the MR-instrument for GCKR, a protein prioritized for both AAM and ANM, with a predominance of associations with lipid and glucose metabolism traits, and blood count traits.”

We have commented on the results of this analysis in the discussion:

Lines 270-275: “Finally, GCKR, the glucokinase regulatory protein, which was associated with both AAM and ANM in our MR and colocalization analyses, exert effects on glucose homeostasis, as demonstrated equally by our Phenoscanner search. Nevertheless, the cis-pQTL of this protein appears to be the most pleiotropic among the MR instruments of all candidate proteins, and therefore the link of this protein with AAM and ANM should be interpreted with caution.”

Lines 296 – 300: “While our Phenoscanner analysis identified associations of the MR instruments of the candidate proteins with BMI and other anthropometric traits, it is not clear if these associations represent pleiotropic pathways, since changes in these traits can be epiphenomena of variations in pubertal or menopausal timing and not the cause of such variations”.

Additional authors' comments.

In view of new results obtained, following the use of the SuSiE plug-in for colocalization (suggested by Reviewer #1), we have added the following information to the manuscript:

Lines 106 – 110 (results section):

“Using the Sum of Single Effects (SuSiE) plug-in in colocalization²⁴, we found that 8 additional proteins (NEGR1, LHB, DLK1, ANXA2, YWHAB, DNAJB12, RMDN1 and HPGDS) colocalized with AAM analysis with an $H_4 > 80\%$ if we considered multiple causal variants within a 1Mb region of each protein's cis-pQTL (Supplementary Table 4A.2).”

Lines 132 – 135 (results section):

“Using colocalization with SuSiE²⁴, we found that 3 additional proteins (LCT, ARL and PLXNA1) colocalized with ANM analysis with a posterior probability $H_4 > 0.8$ if we considered multiple causal variants within a 1Mb region of each protein's cis-pQTL (Supplementary Table 4A.2) (Supplementary Table 4B.2).”

Lines 190 – 193 (discussion section):

“All proteins with positive evidence for colocalization ($H_4 > 0.8$ using colocalization with or without the SuSiE plug-in), or in total 13 proteins for AAM and 7 proteins for ANM, were retained for downstream pathway and enrichment analyses, to further explore the function of the candidate proteins.”

Lines 220 – 224 (discussion section):

“Our MR results were validated using colocalization, and provide evidence for an association between 13 circulating proteins (GCKR, FOXO3, SEMA3G, PATE4, AZGP1, NEGR1, LHB, DLK1, ANXA2, YWHAB, DNAJB12, RMDN1 and

HPGDS) with AAM, and between 7 circulating proteins (CPNE1, TYMP, DNER, ADAMTS13, LCT, ARL and PLXNA1) and ANM.”

Lines 428 – 431 (methods section):

“As such, we performed colocalization with the SuSiE (Sum of Single Effects) plug-in in the coloc R package²⁴, which allows to relax the assumption of a single shared causal variant. To do this, an LD matrix was created using the 1000Genomes phase 3 reference.”

As suggested by reviewer #3, we performed a reverse MR which made the following add to the manuscript:

Lines 144 – 162 (results section):

“In order to test the presence of reverse causation in the association of the candidate proteins for AAM and ANM, we performed reverse MR studies, using AAM or ANM as exposures and the candidate proteins as outcomes. For these analyses, we used 172 and 193 genome-wide significant (GWAS p-value < 5 x 10⁻⁸) and independent (LD R²<0.001) SNPs as instruments for AAM and ANM respectively, which we retrieved from the same REPROGEN Consortium GWAS described above. Our reverse MR analyses were restricted to proteins with cis-pQTLs identified in Sun et al, Suhre et al, and Ferkingstad et al, since full summary-level results of these GWAS were available. We tested reverse MRs for 20 out of the 22 candidate proteins for AAM and for all 19 candidate proteins for ANM. The results of these analyses appear in Supplementary Tables 5A (for AAM) and 5B (for ANM). For each reverse MR analysis, we computed MR estimates using the inverse variance weighted (IVW) method, and three other pleiotropy-robust methods (MR-Egger, weighted median and weighted mode). For ANM, our reverse MR showed evidence of association (MR p-values <0.05) in one out of the four MR methods for 6 proteins (DLK1, DBAJB12, GCKR, NEGR1, TXNDC15, MST1), but for LHB we found stronger evidence of a reverse causal effect of AAM on the level of this protein, with significant results in three out of the four MR methods (Supplementary Table S5A). For ANM we found weak evidence of reverse causation for PNP and TXNDC15, with only one of the four MR methods obtaining estimates with nominally significant p-values (Supplementary Table S5B).”

Lines 224 – 225 (discussion section):

“Our reverse MR analysis showed strong evidence of reverse causation for LHB, meaning that AAM causes altered levels of this protein and not the opposite.”

Lines 276 – 285 (discussion section):

“In order to verify the direction of causality of our MR-prioritized proteins, we conducted reverse MR analyses, with AAM or ANM as exposures and the levels of circulating proteins as outcomes. These analyses confirmed that the direction of the causal association of our forward MR analyses (where proteins were the exposure and AAM or ANM were the outcomes) was right, except for the LHB protein which showed reverse causality with AAM in three MR methods. LHB, the luteinizing hormone (LH) B subunit, is a gonadotropin secreted by the pituitary following pubertal start, which regulates the reproductive function by controlling the production of sex steroid hormones such as estrogen. Therefore, this finding makes sense, since LHB production is the epiphenomenon of pubertal start and not its cause, and can serve as a positive control for our MR study.”

In view of the new analyses, the figures and the supplementary tables have been modified as follows:

Figure 1: Flowchart with the study design. We added the colocalization with SuSiE plug-in, the reverse MR and we adjusted the number of prioritized proteins for the follow-up analyses.

Figure 3: Pathway analysis using GeneMANIA on the colocalization-prioritized proteins. We added the colocalization-prioritized proteins found with the SuSiE plug-in (3A for AAM and 3B for ANM).

Figure 4: Results of gene set enrichment analysis (GTex v8) using FUMA on the colocalization-prioritized proteins. We added the colocalization-prioritized proteins found with the SuSiE plug-in (4A for AAM and 4B for ANM).

Figure 5: Venn diagram summarizing the results of our study for AAM (A) and ANM (B). We added the colocalization-prioritized proteins found with the SuSiE plug-in.

Supplementary Table 4: We added the result of the colocalization using SuSiE plug-in (4A.2 for AAM and 4B.2 for ANM).

Supplementary Table 5: Results of the reverse MR (5A for AAM and 5B for ANM).

Supplementary Table 8: Pathway analysis using GeneMANIA on the colocalization-prioritized proteins. We added the colocalization-prioritized proteins found with the SuSiE plug-in (8A for AAM and 8B for ANM).

Supplementary Table 9: Enrichment analysis using Metascape on the colocalization-prioritized proteins. We added the colocalization-prioritized proteins found with the SuSiE plug-in (9A for AAM and 9B for ANM).

Supplementary Table 10: Results of gene set enrichment analysis (GTEX v8) using FUMA on the colocalization-prioritized proteins. We added the colocalization-prioritized proteins found with the SuSiE plug-in (10A for AAM and 10B for ANM).

Reviewers' comments:

Reviewer #1 (Remarks to the Author):

I would like to commend the authors on their attentiveness to my comments on the previous version of their manuscript. I believe the paper to be much improved and have just a couple of additional comments:

- 1) The results of the Phenoscanner analysis reveal widespread associations between the pQTLs and anthropometric traits in particular. While this is perhaps unsurprising, and could represent epiphenomena, the potential for pleiotropy or mediation via these traits cannot be ruled out. It is true that it is difficult to disentangle these situations with the availability of single pQTLs. Given this uncertainty, can the same conclusions be drawn from the study, particularly regarding the identification of protein targets for treatment of women at risk of extreme AAM or ANM?
- 2) Related to the suggestion that the proteins may represent targets for treatment of women at risk of extreme AAM/ANM, how might these individual be identified and are there any examples of such treatments/therapeutics already on the market?
- 3) The Results for HyPrColoc at GCKR should be shown, along with some interpretation for the lack of simultaneous colocalization with AAM and ANM.

Reviewer #2 (Remarks to the Author):

These responses are mostly clear. I only have remaining comments about the power calculation.

It looks like the power calculation (based on reference 23, Appendix equation A9) takes into account the sample size, proportion of variance in the protein accounted for by the SNP, variance of the protein and the outcome, and beta of the outcome-protein regression. The 2nd item in this list was referred to in my comment last time, and now the 4th item needs more discussion.

Here the proportion of variance in the protein accounted for by the SNP is given as R^2 in Table S11, column D. This is now appropriately included as the R^2 for that individual protein. (Several of the proteins are listed twice. This is because the protein was present in more than one GWAS source. It would be helpful to mention that the R^2 is from Supp Table 3A-3B, column G or provide the GWAS name.)

The other interesting component of this calculation is the beta of the outcome-protein regression. Here the authors used the minimum, maximum, and mean of the betas from the identified proteins from Table S2. Unsurprisingly, the power range is often from 0-100%. This is not informative. The beta from one protein has no bearing on the beta for another; there is no reason to treat this range as any sort of confidence interval. The power for the *mean* of these betas also has no real world meaning whatsoever as it's simply the mean MR coefficient of whatever proteins happened to be in the original assay. (Not to mention that as a mean, it is highly influenced by a couple of outlier proteins with a large effect - yielding high "power3" column for almost everything.) The *actual* beta for each of these rows is available as the beta from each row of (S2A-B, column H) and should be used in the power calculation for each corresponding line of Table S11.

The goal here is to present the estimated posthoc power to detect the observed result for that specific test.

Reviewer #4 (Remarks to the Author):

Re: potential confounding and pleiotropy

As the authors stated in the manuscript, BMI is a known risk factor for both AAM and ANM. Extra analyses are needed for the interpretation of the results. Specially, MVMR should still be considered as the results would less likely be biased by either confounding or pleiotropy due to BMI.

One reviewer stated that it requires MORE instruments than the risk factors to perform MVMR; however, the exact statement in the paper (PMC4325677) provided by the same reviewer is "...we suppose there are multiple genetic variants (AT LEAST AS MANY VARIANTS AS THERE ARE RISK FACTORS) which have different magnitudes of effect on the risk factors. These genetic variants can be used to estimate the causal effects of each risk factor even if none of the variants are specifically associated with any 1 particular risk factor." This means that authors can still perform MVMR using one SNP associated with protein levels and one (or more) SNP associated with BMI as long as the SNP effects on the protein and BMI are different.

Another reviewer queried about the mediation versus pleiotropy, this can be tested using MVMR using different models mentioned in the same paper (PMC4325677). Specifically, the model in Figure 3B fits a causal path between two exposures, which takes into account the mediation effect of protein on AAM/ANM via BMI (or vice versa).

Given that a high percentage of instruments are associated with anthropometric traits, in addition to the MVMR including BMI as an extra exposure, the authors could provide a list of proteins whose pQTLs are NOT associated with anthropometric traits because we should be more confident for the MR results of these proteins.

Re: Reverse causality

In addition to the reverse MR, the authors should also perform MR Steiger test, which also informs the potential presence of reverse causality.

Lastly, I would like to request the authors to include the SNP effects (effect allele, non-effect allele, beta, se, p-value) on all exposures (proteins) and outcomes (AAM and ANM) in Supplementary Table 2A and 2B, as well as their effects on BMI. This will help readers to replicate findings from this work and also inform potential confounding and pleiotropy as mentioned above.

Reviewer #1 (Remarks to the Author):

I would like to commend the authors on their attentiveness to my comments on the previous version of their manuscript. I believe the paper to be much improved and have just a couple of additional comments:

1) The results of the Phenoscanner analysis reveal widespread associations between the pQTLs and anthropometric traits in particular. While this is perhaps unsurprising, and could represent epiphenomena, the potential for pleiotropy or mediation via these traits cannot be ruled out. It is true that it is difficult to disentangle these situations with the availability of single pQTLs. Given this uncertainty, can the same conclusions be drawn from the study, particularly regarding the identification of protein targets for treatment of women at risk of extreme AAM or ANM?

We agree with the reviewer that pQTLs for many proteins exert pleiotropic effects. We have attempted to classify the pQTLs into those with high, medium and low evidence of pleiotropy based on our Phenoscanner search (Supplementary Tables 7A and 7B), and based on these results, we have modified results in discussion sections (lines 178 – 182 and 256-258). Also, we have added a two-step network MR analysis to explore the presence of mediation by the BMI (lines 185 - 201).

2) Related to the suggestion that the proteins may represent targets for treatment of women at risk of extreme AAM/ANM, how might these individuals be identified and are there any examples of such treatments/therapeutics already on the market?

We thank the reviewer for this insightful comment. We have performed a search in the OpenTargets platform to identify which of the genes related to the candidate proteins are drug targets. The results appear in the results section of the manuscript (lines 242 - 245), and in Supplemental Table 13. Among all candidate proteins, only 4 protein-biomarkers for ANM are encoded by genes which are targets for existing drugs (ADH5, PNP, TNFRSF17, TYMP). We are also commenting on this in the discussion section (lines 360-362)

3) The Results for HyPrColoc at GCKR should be shown, along with some interpretation for the lack of simultaneous colocalization with AAM and ANM.

We have added the results of the HyPrColoc in Supplemental Table 5C.

Reviewer #2 (Remarks to the Author):

These responses are mostly clear. I only have remaining comments about the power calculation.

It looks like the power calculation (based on reference 23, Appendix equation A9) takes into account the sample size, proportion of variance in the protein accounted for by the SNP, variance of the protein and the outcome, and beta of the outcome-protein regression. The 2nd item in this list was referred to in my comment last time, and now the 4th item needs more discussion.

Here the proportion of variance in the protein accounted for by the SNP is given as R2 in Table S11, column D. This is now appropriately included as the R2 for that individual protein. (Several of the proteins are listed twice. This is because the protein was present in more than one GWAS source. It would be helpful to mention that the R2 is from Supp Table 3A-3B, column G or provide the GWAS name.)

Following the reviewers' comment, we have specified in the legend of Supplemental Table 4A and 4B that the R2 estimates are the same as those in Supplemental Tables 3A and 3B.

The other interesting component of this calculation is the beta of the outcome-protein regression. Here the authors used the minimum, maximum, and mean of the betas from the identified proteins from Table S2. Unsurprisingly, the power range is often from 0-100%. This is not informative. The beta from one protein has no bearing on the beta for another; there is no reason to treat this range as any sort of confidence interval. The power for the *mean* of these betas also has no real world meaning whatsoever as it's simply the mean MR coefficient of whatever proteins happened to be in the original assay. (Not to mention that as a mean, it is highly influenced by a couple of outlier proteins with a large effect - yielding high "power3" column for almost everything.) The *actual* beta for each of these rows is available as the beta from each row of (S2A-B, column H) and should be used in the power calculation for each corresponding line of Table S11. The goal here is to present the estimated posthoc power to detect the observed result for that specific test.

We agree with the reviewer's comment. We have now replaced the beta for the power analysis with the actual MR beta from Supplemental Tables 2A and 2B. We provide all the available MR betas for each protein for proteins present in more than one proteomic GWAS - in some cases the power was less than 80%, which explains the non-significant MR p-value for these proteins when using instruments from those GWAS.

Reviewer #4 (Remarks to the Author):

Re: potential confounding and pleiotropy

As the authors stated in the manuscript, BMI is a known risk factor for both AAM and ANM. Extra analyses are needed for the interpretation of the results. Specially, MVMR should still be considered as the results would less likely be biased by either confounding or pleiotropy due to BMI.

One reviewer stated that it requires MORE instruments than the risk factors to perform MVMR; however, the exact statement in the paper (PMC4325677) provided by the same reviewer is "...we suppose there are multiple genetic variants (AT LEAST AS MANY VARIANTS AS THERE ARE RISK FACTORS) which have different magnitudes of effect on the risk factors. These genetic variants can be used to estimate the causal effects of each risk factor even if none of the variants are specifically associated with any 1 particular risk factor." This means that authors can still perform MVMR using one SNP associated with protein levels and one (or more) SNP associated with BMI as long as the SNP effects on the protein and BMI are different.

Another reviewer queried about the mediation versus pleiotropy, this can be tested using MVMR using different models mentioned in the same paper (PMC4325677). Specifically, the model in Figure 3B fits a causal path between two exposures, which takes into account the mediation effect of protein on AAM/ANM via BMI (or vice versa).

We have looked more carefully into the possibility of performing MVMR with one SNP-instrument.

Specifically, we found the following mention in the paper by Sanderson et al (PMID: 30535378). Page 4: « "Important considerations": To conduct an MVMR analysis, it is necessary to have at least as many genetic instruments as there are exposures to be instrumented in the model; this is true regardless of whether single-sample or two-sample summary data are used. » Also, we performed a test in the MVMR package using example data:

Considering 3 exposures:

- If only 1 SNP per exposure taken into account: function returns a NA
- If 2 SNPs per exposure taken into account: function returns a NA
- If 3 SNPs per exposure taken into account: function returns the estimated betas but no test carried out (no SE, T-value, P-value)
- If 4 SNPs per exposure taken into account: function returns betas, SE, T-value and P-value)

In our case, we would therefore need at least 3 SNPs to consider 2 exposures but for each protein we only have to consider a single instrument (SNP).

In order to take the mediating effect of BMI into account in our analysis, we performed a two-step network MR method. The results can be seen in lines 185 - 201 as well as in Supplementary Tables 8A and 8B. We were able to demonstrate that the effects of the NEGR1 and AZGP1 proteins were mediated by BMI.

Given that a high percentage of instruments are associated with anthropometric traits, in addition to the MVMR including BMI as an extra exposure, the authors could provide a list of proteins whose pQTLs are NOT associated with anthropometric traits because we should be more confident for the MR results of these proteins.

We agree with the reviewer that more information should be given on the results of our search for pleiotropic effects

in Phenoscanner. We have classified the pQTLs as those with high, medium and low evidence of pleiotropy, based on the percentage of pleiotropic associations (Supp Tables 7A and 7B).

Re: Reverse causality

In addition to the reverse MR, the authors should also perform MR Steiger test, which also informs the potential presence of reverse causality.

As suggested, we performed the Steiger test in our significant MR associations. The results appear in lines 144 – 146 and in Supplementary Table 6AB.2.

Lastly, I would like to request the authors to include the SNP effects (effect allele, non-effect allele, beta, se, p-value) on all exposures (proteins) and outcomes (AAM and ANM) in Supplementary Table 2A and 2B, as well as their effects on BMI. This will help readers to replicate findings from this work and also inform potential confounding and pleiotropy as mentioned above.

The information was added in Supplemental Tables 2A and 2B.

Reviewers' comments:

Reviewer #2 (Remarks to the Author):

The comments have been fully addressed.

Reviewer #4 (Remarks to the Author):

Re: number of genetic instruments required for performing MVMR

The paper (Sanderson et al) the authors brought up also says "to have as many genetic instruments as there are exposures to be instrumented in the model...", which indicates for two exposures (i.e. one protein and BMI) you just need a minimum of two genetic instruments.

According to the MR Dictionary (<https://mr-dictionary.mrcieu.ac.uk/term/multivariable/>), you would want to use SNPs that are associated both exposures to be included in the MVMR. If the instruments you used in the test were only associated with one exposure, it would be the reason why the 2-sample MR package returns an error message or NA. Could you please give it a try using SNPs that are associated with both protein and BMI in MVMR?

Reviewer #4 (Remarks to the Author):

Re: number of genetic instruments required for performing MVMR

The paper (Sanderson et al) the authors brought up also says "to have as many genetic instruments as there are exposures to be instrumented in the model...", which indicates for two exposures (i.e. one protein and BMI) you just need a minimum of two genetic instruments.

According to the MR Dictionary (<https://mr-dictionary.mrcieu.ac.uk/term/multivariable/>), you would want to use SNPs that are associated both exposures to be included in the MVMR. If the instruments you used in the test were only associated with one exposure, it would be the reason why the 2-sample MR package returns an error message or NA. Could you please give it a try using SNPs that are associated with both protein and BMI in MVMR?

Following the reviewer's suggestion, we conducted an MVMR analysis for each of the candidate proteins for AAM or ANM using their respective *cis*-pQTL and SNP-instruments for either pediatric or adult BMI, and extracting effects of these SNPs from both the protein and BMI GWASes. This limited the number of proteins for which we could perform MVMR to those with available full summary level GWAS data. The GWAS for pediatric or adult BMI were the same as those used in our two-step network MR analysis. The manuscript has been updated to accommodate this analysis in the results and discussion sections and we added two new Supplementary Tables (Supplementary Tables 9A and 9B).

Results Section: **Lines 201-212** : "In addition, we tested the simultaneous effects of BMI and proteins on AAM or ANM by conducting multivariable MR (MVMR- see Supplementary Tables 9A and 9B). To do this, for each protein, we used its *cis*-pQTL and SNP-instruments for BMI (pediatric or adult), and retrieved their effects on both exposures and on AAM or ANM. For the ANM MVMR, we used 18 SNPs, 1 *cis*-pQTL for each protein and 17 independent genome-wide significant SNPs for pediatric BMI. For ANM, we used 522 SNPs, 1 *cis*-pQTL for each protein and 521 independent SNPs from the BMI GWAS. Our MVMR analyses for AAM showed that pediatric BMI had significant effects on AAM, after considering the simultaneous effect of each of 13 candidate proteins. Contrarily the MR effect of all 13 proteins was non-significant after accounting for pediatric BMI. In the MVMR analyses for ANM, the effect of adult BMI was consistently not significant, but two proteins (ADAMTS13 and CPNE1) maintained a significant effect on ANM after accounting for adult BMI."

Discussion Section: **Lines 269-271**: "Our MVMR analyses also suggest a possible mediating or confounding effect of BMI in the association of candidate proteins and AAM, and the same applied for most candidate proteins for ANM, with the exception of ADAMTS13 and CPNE1."

REVIEWERS' COMMENTS:

Reviewer #4 (Remarks to the Author):

The authors have fully addressed my comments.